# The Feasibility of Utilizing Cultured *Cordyceps militaris* Residues in Cosmetics: Biological Activity Assessment of Their Crude Extracts

**DOI:** 10.3390/jof7110973

**Published:** 2021-11-16

**Authors:** Punyawatt Pintathong, Putarak Chomnunti, Sarita Sangthong, Areeya Jirarat, Phanuphong Chaiwut

**Affiliations:** 1Green Cosmetic Technology Research Group, Mae Fah Luang University, Chiang Rai 57100, Thailand; sarita.san@mfu.ac.th (S.S.); areeyajirarat@gmail.com (A.J.); phanuphong@mfu.ac.th (P.C.); 2School of Cosmetic Science, Mae Fah Luang University, Chiang Rai 57100, Thailand; 3School of Science, Mae Fah Luang University, Chiang Rai 57100, Thailand; putarak.cho@mfu.ac.th; 4Center of Excellence in Fungal Research, Mae Fah Luang University, Chiang Rai 57100, Thailand

**Keywords:** *Cordyceps militaris*, solid residue, phenolic compound, antioxidant activity, tyrosinase inhibitory activity, UV absorption, cell cytotoxicity, cosmetics

## Abstract

Solid-based residues (SBRs) left from harvesting the fruiting bodies of cultured *Cordyceps* mushrooms are a challenge to sustainability. Therefore, in this study, the SBRs from the cultivation of *Cordyceps militaris* (*C. militaris*) via solid-state fermentation (SSF) were employed to prepare crude extracts, with the aim of considering their possible use in cosmetics. The SBRs obtained from cultivation with solid media containing defatted rice bran mixed with barley, white rice, Riceberry rice, and wheat were named SBR-B, SBR-R, SBR-Rb, and SRB-W, respectively. They were extracted with solvents of differing polarity and then evaluated for their total phenolic content (TPC), total flavonoid content (TFC), and total carbohydrate content (TCC). In addition, antioxidant and tyrosinase inhibitory activities, photoprotection, and cytotoxicity were also assessed. The results revealed that the total bioactive contents and biological capacities of crude SBR extracts were significantly influenced by the types of SBR and extraction solvent (*p* < 0.05). The SBR-B extracted with hot water exhibited the highest antioxidant activity (66.62 ± 2.10, 212.00 ± 3.43, and 101.62 ± 4.42 mg TEAC/g extract) when assayed by DPPH, ABTS, and FRAP methods, respectively, whereas tyrosinase inhibitory activity (51.13 ± 1.11 mg KAE/g extract) with 90.43 ± 1.96% inhibition at 1 mg/mL was excellently achieved by SBR-Rb extracted by 50% (*v*/*v*) ethanol. Correlations between bioactive contents in the crude extracts and their biological activities were mostly proven at a strong level (*p* < 0.01). The capability of the crude extracts to absorb UV over the range of 290–330 nm disclosed their potential roles as natural UV absorbers and boosters. Cytotoxicity analysis using fibroblast cell lines tested with hot water and 50% (*v*/*v*) ethanolic SBR extracts demonstrated safe use within a concentration range of 0.001–10 mg/mL. Interestingly, their fibroblast proliferative capacity, indicating anti-aging properties, was highly promoted. The chemical composition analyzed via LC–MS/MS techniques showed that seven phenolic acids and four flavonoids were identified in the crude SBR extracts. Furthermore, the other compounds present included nucleosides, nucleobases, amino acids, sugars, phospholipids, alkaloids, organic acids, vitamins, and peptides. Therefore, it is emphasized that SBRs from *C. militaris* can be a prospective source for preparing crude extracts employed in cosmetics. Lastly, they could be further utilized as multifunctional ingredients in cosmetics and cosmeceuticals.

## 1. Introduction

Many species of *Cordyceps* are widely recognized as medicinal or edible mushrooms that have been used for a long time to improve health and wellbeing, such as against cancer, for immune system enhancement, and for combating age-related diseases. They are ascomycetous fungi that grow as endoparasites on arthropods. The pharmacological actions of the *Cordyceps* mushrooms have been widely examined and evaluated for antioxidant, anti-inflammatory, immunomodulatory, antitumor, anti-aging, and neuroprotective effects [1,2]. In addition, numerous studies have shown their medicinal properties for the development of drugs and health care products. According to a market analysis report [3], the global market size of *Ophiocordyceps sinensis* (formerly known as *C. sinensis*) and *C. militaris* was valued at USD 473.4 million in 2018, and the market demand has been predicted to grow by 10.4% CAGR (compound annual growth rate) from 2018 to 2026.

The fungus *C. militaris* belongs to the family Cordycipitaceae; it grows as a parasite in *Lepidopteran pupa*, and the color of its fruiting bodies is yellow or orange. The mycelium and fruiting body of the *C. militaris* are widely used in the pharmaceutical and health care industries, as medical materials, dietary supplements, and food additives, based on historical and research evidence. The potential components found in the mycelia and fruiting bodies are cordycepin [4], polysaccharides [5], proteins [6], and phenolic compounds [7], which can be extremely effective based on their bioactivity. Moreover, these substances have also been reported to significantly enhance oxygen mobilization, improve ATP synthesis, stabilize blood sugar metabolism, and actualize DNA repair; thus, these mechanisms could help to reduce the risk of disease and strengthen human health [1]. Consequently, marketing requirements are continuously increasing; while the *C. militaris* host is specialized and rare in nature, it develops quite slowly, grows only in specific places, and has a small size [8]. Artificial cultivation has been extensively developed for large-scale production, and also ensures the quality of chemical constituents in cultured *Cordyceps* products.

Today, many manufacturers have commercially developed *Cordyceps* products from the mycelium and fruiting body of *C. militaris* grown on artificial media using optimized cultivation methods. Research data have shown that several compounds produced from its cultivation—including cordycepin, amino acids, proteins, polysaccharides, nucleosides, mannitol, carotenoids, and trace minerals—are similar to the products derived from natural sources [9]. The fruiting bodies of *C. militaris* are mostly used and prepared by cultivation via SSF. Various types of solid media containing primary carbon sources can be employed—for example, rice, cereal grains, and insects (such as silkworm pupae) [9,10]. At the end of the cultivation process, after harvesting the fruiting bodies, the remaining large amount of solid medium and mycelium waste might lead to environmental and waste management problems. However, many researchers have attempted to solve these problems. Ni et al. [11] developed a new column chromatographic extraction (CCE) method for the extraction, separation, and purification of pure cordycepin from the solid medium remaining from the industrial cultivation of *C. militaris*.

Although bioactive compounds extracted from *C. militaris* are preferably used in healthy foods and dietary supplements, their application in cosmetics and cosmeceuticals rarely emerges. Therefore, in the present study, the solid-based residues (SBRs) left from solid-state cultivation of *C. militaris* were prepared as a material source for the preparation of crude extracts. The crude extracts were investigated for bioactive compounds and biological activities concerning cosmetics. The possible use of these bioactive compounds as promising active ingredients in cosmetics was also assessed.

## 2. Materials and Methods

### 2.1. Plant Materials and Chemicals

Defatted rice bran extruded into pellet form was obtained from Thai Edible Oil Co., Ltd., Thailand. White (jasmine) rice and Riceberry rice were purchased from a local market, while barley and wheat were purchased via online marketplaces. Folin–Ciocalteu phenol reagent was obtained from Loba Chemie (Colaba, Mumbai, India). DPPH (2,2-diphenyl-1-picrylhydrazyl), ABTS (2,2′-azino-bis (3-ethylbenzothiazoline-6-sulfonic acid), TPTZ (2,4,6-Tris(2-pyridyl)-s-triazine), L-DOPA (3,4 dihydroxy-L-phenylalanine), Trolox ((±)-6-hydroxy-2,5,7,8-tetramethylchromane-2-carboxylic acid), gallic acid, quercetin, kojic acid, and mushroom tyrosinase (EC 1.14.18.1) were purchased from Sigma-Aldrich (St. Louis, MO, USA). All phenolic standards of LC–MS grade were purchased from Sigma-Aldrich (St. Louis, MO, USA). Acetonitrile and water for LC–MS analysis were bought from J.T. Baker (Radnor, PA, USA). Formic acid was of LC–MS grade and was obtained from Thermo Fisher Scientific (Waltham, MA, USA). All other reagents were either of analytical grade or of the highest quality available.

### 2.2. Fungal Strain

The *C. militaris* strain P-1-012 was obtained from the Engineering Research Center of Southwest Bio-Pharmaceutical Resources, Ministry of Education, Guizhou University, China. It was cultivated on potato dextrose agar (PDA) and incubated for 7–10 days for maximum growth. To prepare the seed culture, small pieces of mycelium were transferred to a 500 mL Erlenmeyer flask containing 100 mL of potato dextrose broth (PDB) and incubated in an incubator shaker at 18 °C and 150 rpm with 80% humidity for 7 days.

### 2.3. Cultivation of C. militaris by SSF

SSF was conducted in a 355 mL glass bottle with a lid. In this study, four formulae of solid media containing defatted rice bran and different types of cereal—i.e., barley, white rice, Riceberry rice, and wheat—were prepared. Twenty grams of each solid medium (10 g of each component) mixed with DI water (50 mL) were sterilized by autoclaving at 121 °C for 15 min. A volume of 15 mL of seed culture was transferred to a solid medium and incubated at 20 °C under dark conditions for 7 days. The cultures were maintained at 20 °C during the day (the white light), with relative humidity (RH) at 80%, for 45 days. Growth characteristics in batch culture and the height of each fruiting body and mycelium were measured. After harvesting the fruiting bodies, the solid-based residues (SBRs) left after SSF—containing both mycelium and the remains of the medium—were collected for further study.

### 2.4. Preparation of Solid-Based Residues (SBRs)

The SBRs obtained from different solid media were weighed and dried at 45 ± 1 °C for 2 days to obtain a constant weight. The dry biomass was ground into a fine powder using a cooking blender, collected, and stored in a zip bag at −20 °C until use. The SBRs were coded as SBR-B, SBR-R, SBR-Rb, and SBR-W, as obtained from media containing barley, white rice, Riceberry rice, and wheat, respectively.

### 2.5. Preparation of Crude Extracts

The powdered samples of SBRs were extracted via the solid–liquid extraction method, using different solvents, including ethyl acetate, acetone, ethanol, 50% (*v*/*v*) acetone, 50% (*v*/*v*) ethanol, water, and hot water. A 15 g sample was mixed with a solvent (150 mL) at a solid–liquid ratio of 1:10 *w*/*v*. The mixture was shaken on an orbital shaker at 125 rpm under ambient temperature for 3 h. For hot water extraction, the procedure was carried out in an incubator shaker at 125 rpm and 95 ± 1 °C for 3 h. The experiment was conducted in triplicate. The supernatant was obtained using a vacuum filtration apparatus through Whatman filter paper no.1. The solvent was removed using a rotary evaporator and/or freeze-dryer, and the crude extract obtained was calculated for extraction yield, and then collected and stored at −20 °C until use.

### 2.6. Proximate Analysis

Proximate analysis of the dried SBRs was carried out for moisture, ash, fiber, lipid, protein, and carbohydrate contents, using the standard methods of the Association of Official Analytical Chemists (AOAC) [12]. All results were calculated in terms of the percentage of dry weight (%dw). Moisture content was measured using a hot-air oven at a temperature of 105 ± 1 °C. For ash content, the sample was incinerated in a furnace at 600 °C. Crude protein content was measured based on the Kjeldahl method, using a nitrogen combustion instrument (FP528, LECO, USA). The percentage of crude protein was calculated as % total nitrogen × 6.25. Lipid content was detected using a fat-extraction system (Soxtec 8000, FOSS, Denmark), whereas fiber content was detected using a fiber analyzer (Fibertec8000, FOSS, Denmark). Total carbohydrate content was calculated using the following equation:Carbohydrate content (%) = 100 − (% moisture content + % ash content + % total protein + % fiber content + % total fat content)(1)

### 2.7. Determination of Cordycepin

The cordycepin content of the dried samples of fruiting bodies and SBRs, extracted with 85 °C water for 2.5 h and ultrasonication for 30 min, were quantitatively determined via high-performance liquid chromatography (HPLC), using an Agilent 1260 Infinity system and a reversed-phase column (Agilent ZORBAX Eclipse XDB-C18, 4.6 × 150 mm, 5 µm). A 10 µL aqueous solution of each sample, filtered through a microporous membrane (0.45 µm), was injected. The separation was carried out using 2% acetonitrile–water (0.5% trifluoroacetic in water) as the mobile phase, with a flow rate of 1 mL/min. The UV detector was set to 260 nm. The standard cordycepin was also used to prepare the calibration curve, with the linear regression equation (*y* = 4.5429*x* + 1.6447, *R*^2^ = 0.9998). The results were expressed as mg/g of dry sample.

### 2.8. Determination of Extraction Yield

Extraction yield is a measurement of the solvent’s efficiency in extracting specific compounds from original materials. It can be expressed as a percentage of extract (g) per 100 g of dried SBR, calculated using the following equation:Extraction yield (%) = (*W*_1_/*W*_2_) × 100(2)
where *W*_1_ is the weight of the crude extract and *W*_2_ is the weight of the SBR sample.

### 2.9. Determination of Total Phenolic Content (TPC)

Total phenolic content (TPC) was determined using Folin–Ciocalteu reagent according to the method described by Singleton et al. [13], with a slight modification; gallic acid was used as the standard. The crude extract was first diluted with an appropriate solvent. The sample solution was successively mixed with 0.25 mL of Folin–Ciocalteu reagent and 1.5 mL of sodium carbonate solution (7.5% *w*/*v*). The mixture was allowed to stand for 30 min. An absorbance of 765 nm was used to measure the obtained blue-colored mixture with a spectrophotometer. A calibration curve was prepared using gallic acid as a standard, and the results were expressed as milligrams of gallic acid equivalents per gram of crude extract (mg GAE/g extract).

### 2.10. Determination of Total Flavonoid Content (TFC)

Total flavonoid content (TFC) was determined via the aluminum colorimetric assay, following the method of Kumar et al. [14], with a slight modification, and quercetin was used as a standard. The proper volume of sample solution was mixed with 0.15 mL of 5% (*w*/*v*) NaNO_2_ solution, followed by 10% (*w*/*v*) AlCl_3_ solution (0.15 mL). After incubation at ambient temperature for 5 min, 1 mL of 4% (*w*/*v*) NaOH solution was added, and the mixture was incubated again for 5 min. Next, the developed color of the mixture was measured at 510 nm using a spectrophotometer. The result was expressed as milligrams of quercetin equivalents per gram of extract (mg QE/g extract).

### 2.11. Determination of Total Carbohydrate Content (TCC)

Total carbohydrate content was determined via a phenol–sulfuric acid assay, using glucose as a standard [15]. The extract solution was mixed with 5% (*w*/*v*) phenol solution, followed by sulfuric acid. The mixture was incubated at 50 °C for 10 min, and then cooled in an ice bath for 10 min. The absorbance was read at 490 nm against the blank. The result was expressed as milligrams of glucose equivalents per gram of extract (mg GE/g extract).

### 2.12. Determination of Antioxidant Activity

#### 2.12.1. DPPH Radical Scavenging Activity

The DPPH radical scavenging activity of the samples was determined according to the method of Thaipong et al. [16], with some modifications. An aliquot of each sample was first dissolved in ethanol, and then mixed with 0.1 mM DPPH reagent. The mixture was incubated in the dark for 30 min. Discoloration of the mixture was measured at 517 nm using a spectrophotometer. The percentage of DPPH radical inhibition was calculated using the following equation:Free radical inhibition (%) = ((Ac − As)/Ac) × 100(3)
where Ac is the absorbance of the control and As is the absorbance of the sample.

Trolox was used as a standard, and the result was expressed as milligrams of Trolox equivalent antioxidant capacity per gram of extract (mg TEAC/g extract).

#### 2.12.2. ABTS Radical Scavenging Activity

The ABTS radical cation scavenging activity of samples was determined according to the method of Re et al. [17], with slight modifications. An aliquot of each sample was first diluted with 50 mM phosphate buffer (pH 7.4) to a volume of 1 mL. Then, the mixture was mixed with ABTS reagent (2 mL) and incubated under dark conditions for 30 min. The blue-decolorized reaction was measured at 734 nm, and a decrement of absorbance was calculated to radical inhibition using Equation (3), as mentioned above. Trolox was used as a standard. The scavenging capacity was expressed as milligrams of Trolox equivalent antioxidant capacity per gram of extract (mg TEAC/g extract).

#### 2.12.3. Ferric-Reducing Antioxidant Power (FRAP)

The reducing power of samples was determined via the Fe(II)–TPTZ complex method using Trolox as a standard, as described by Benzie and Strain [18], with slight modifications. One milliliter of each sample diluted with 0.3 M acetate buffer (pH 3.6) was mixed with 2 mL of FRAP reagent. The reaction was initiated via incubation at 37 °C for 30 min in a water bath. The appearance of a blue-colored solution was measured at 593 nm using a spectrophotometer. The result was expressed as milligrams of Trolox equivalent antioxidant capacity (TEAC) per gram of extract (mg TEAC/g extract).

### 2.13. Determination of Tyrosinase Inhibitory Activity

Inhibition of the tyrosinase enzyme was investigated via the dopachrome method as previously described [19], with some modifications. L-DOPA was used as a substrate. An aliquot of each sample was diluted with 50 mM phosphate buffer (pH 6.8) to a volume of 1.76 mL, and then mixed with 1000 units/mL of mushroom tyrosinase solution (0.04 mL). The mixture was pre-incubated at 37 °C for 4 min. The enzymatic reaction was initiated by adding 0.2 mL of 15 mM L-DOPA, followed by further incubation for 10 min at 37 °C. The absorbance at 475 nm was measured. The percentage of tyrosinase inhibition was calculated and transformed to tyrosinase inhibitory activity, which was expressed in terms of milligrams of kojic acid equivalents per gram of extract (mg KAE/g extract).

### 2.14. Determination of Photoprotective Activity

UV absorption of the extract was carried out via the scanning method, using a UV–Vis spectrophotometer (Libra S80, Biochrom, Holliston, MA, USA). The final concentration of 0.5 mg/mL of different SBR extracts was prepared using an appropriate solvent. Each extract solution was scanned in the range of 290–400 nm, using the same solvent as a blank.

### 2.15. Determination of Cytotoxic Activity

#### 2.15.1. Cell Culture and Treatment

The NIH/3T3 (ATCC CRL-1658) fibroblast cell lines were purchased from the American Type Culture Collection (ATCC, Manassas, VA, USA). Cells were cultured in Dulbecco’s modified Eagle’s medium (DMEM) containing 10% fetal bovine serum (FBS) and 1% (*v*/*v*) penicillin–streptomycin; they were maintained in a humidified atmosphere with 5% CO_2_ at 37 °C.

#### 2.15.2. MTT-Based Cytotoxicity Assay

The in vitro cytotoxicity of the extract was examined by measuring the reduction in soluble MTT to form water-insoluble formazan, as described by Freshney [20], with some modifications. The cultured NIH/3T3 cell lines were seeded at a density of 1 × 10^5^ cells/mL in a 96-well plate containing 180 μL of culture medium and then incubated in a CO_2_ incubator. After 24 h, the cells were treated with 20 μL of various concentrations (0.001–10 mg/mL) of extraction solutions, and then incubated for a further 48 h. A volume of 25 μL of 5 mg/mL of MTT was then added and incubated for another 4 h. The medium was discarded, and the purple MTT–formazan crystals were dissolved with 100 μL of DMSO and 25 μL of Sorensen’s glycine buffer. After 5 min, absorbance at 570 nm was measured using a microplate reader. The experiments were performed in triplicate. The percentage of cell viability was calculated using the following equation:% Cell viability = (As/Ac) × 100(4)
where As is the absorbance of extract solution and Ac is the absorbance of the control (solvent).

### 2.16. LC–MS/MS Analysis

#### 2.16.1. LC-ESI-QqQ-MS/MS Conditions

A liquid chromatography triple-quadrupole mass spectrometer system equipped with electrospray ionization source operating (LC-ESI-QqQ-MS/MS) was used to identify and quantify phenolic compounds in the SBR extract using the Shimadzu UHPLC binary pump Nexera X2 with LCMS-8060 (Shimadzu, Kyoto, Japan). One microliter of sample extract was injected into an ACE C-18-PFP (1.7 μm, 2.1 × 100 mm) column with the oven temperature set to 30 °C. The gradient elution using 0.2% (*v*/*v*) formic acid as mobile phase A and acetonitrile as mobile phase B, with a flow rate of 0.4 mL/min, was employed for the following gradient program: 0.00–0.30 min, 10% (B); 2.40 min, 15% (B); 3.25–3.60 min, 20% (B); 6.2–7.0 min, 95% (B); 7.5–11 min, 10% (B). The ESI source operating was processed in negative ionization mode under the following experimental conditions: interface voltage: 4000 V; DUIS corona needle voltage: 4500 V; interface temperature: 300 °C; DL temperature: 250 °C; heat block temperature: 400 °C; nebulizer gas flow: 3 L/min; heater gas flow: 10 L/min; and drying gas flow: 10 L/min.

LC–MS/MS data were detected in multiple reaction monitoring (MRM) mode and processed using Lab Solutions software. A total of 32 standard phenolic compounds were first used for screening and identification. The MRM transitions from the precursor ion to the product ion for the qualified phenolic compound were selected as follows: gallic acid, *m*/*z* 169.10 > *m*/*z* 125.20 (CE: 17 V); chlorogenic acid, *m*/*z* 352.95 > *m*/*z* 191.05 (CE: 20 V); caffeic acid, *m*/*z* 178.80 > m/z 135.20 (CE: 18 V); *p*-coumaric acid, *m*/*z* 162.85 > *m*/*z* 119.00 (CE: 15 V); *o*-coumaric acid, *m*/*z* 163.00 > *m*/*z* 119.00 (CE: 15 V); protocatechuic acid, *m*/*z* 153.25 > *m*/*z* 109.05 (CE: 17 V); *p*-hydroxybenzoic acid, *m*/*z* 137.05 > *m*/*z* 92.95 (CE: 16 V); rutin, *m*/*z* 609.05 > *m*/*z* 300.20, 271.10 (CE: 37, 55 V); quercetin, *m*/*z* 300.75 > *m*/*z* 151.20, 179.00 (CE: 22, 20 V); apigenin, *m*/*z* 269.10 > *m*/*z* 117.00 (CE: 34 V); and naringenin, *m*/*z* 271.05 > *m*/*z* 151.05, 119.05, 106.95 (CE: 18, 27, 26 V). Quantification results were calculated using standard curves and expressed in μg/g of extract.

#### 2.16.2. LC-ESI-QTOF-MS/MS Conditions

The tentative compounds in the SBR extracts were identified using a liquid chromatography quadrupole time-of-flight mass spectrometer (LC-QTOF-MS)—model G6545B (Agilent, Santa Clara, CA, USA)—equipped with a Dual Agilent Jet Stream–Electrospray Ionization (Dual AJS ESI) source. The sample extract (1 μL) was injected into Poroshell EC C18 (2.7 μm, 2.1×150 mm) column (Agilent, Santa Clara, CA, USA) with the column temperature set to 35 °C. Mobile phase A (0.1% (*v*/*v*) formic acid) and mobile phase B (acetonitrile), with a flow rate at 0.2 mL/min, were employed using a binary pump in the following gradient program: 1 min, 5% (B); 10–13 min, 83% (B); 20–25 min, 100% (B); 27–33 min, 5% (B). Ionization was performed in positive and negative electrospray (ESI) modes; the optimized conditions were set as follows: gas temperature: 300 °C; drying gas: 11.0 L/min; nebulizer: 45 psi; capillary voltage: 3500 V; fragmentor: 135 V; skimmer: 65 V; and collision energy was fixed at 10, 20, and 40 V. The mass spectrometer employed full scan ranges of *m*/*z* 100–1100 for MS, and the reference masses were set at *m*/*z* 121.050873 to 922.009798 in positive ion mode and *m*/*z* 112.985587 to 1033.988109 in negative ion mode. LC-ESI-QTOF-MS data were processed using Agilent Mass Hunter Workstation software (version B.06.00; Agilent Technologies, Santa Clara, CA, USA).

### 2.17. Correlation and Statistical Analysis

All experiments were performed in triplicate, and data were expressed as mean ± standard deviation (SD). The data obtained from the completely randomized design (CRD) experiment were statistically analyzed via one-way analysis of variance (ANOVA), Tukey’s post hoc test (α = 0.05), and correlational analysis using the SPSS software program, version 22.0 (SPSS Inc., Chicago, IL, USA). Differences with *p*-values less than 0.05 (*p* < 0.05) were considered significant.

## 3. Results

### 3.1. Prepartion of Solid-Based Residues (SBRs) Obtained from Solid-State Cultivation

#### 3.1.1. Physical Appearance and Yield

In this study, defatted rice bran—a byproduct of rice bran oil production—was employed as a new alternative solid medium for *C. militaris* cultivation via SSF. Unfortunately, our preliminary study found that it was limited to growing only at the top surface of the defatted rice bran medium. This is because there was no space, air, or surface area inside the solid medium, which was tightly stacked and packed. Therefore, it could not sufficiently support the fully aerobic growth conditions of *C. militaris*. To solve this problem, different cereal grains—including barley, white rice, Riceberry rice, and wheat—were incorporated with the defatted rice bran at a 1:1 (*w*/*w*) ratio to prepare different formulae of solid media. The results showed that it can grow well on all solid media, and the fruiting body was produced within 45 days (Figure 1a–d). This evidence indicates that defatted rice bran can be used as a solid medium for the cultivation of *C. militaris* if cereal grains are combined with it. As observed for 45 days, the highest growth was found in the presence of Riceberry rice, followed by wheat, barley, and white rice (Figure 1a–d, respectively). The highest yield of fruiting bodies was obtained in the presence of Riceberry rice (12.85 ± 0.37% *w*/*w*), followed by wheat (11.61 ± 0.32% *w*/*w*), barley (8.91 ± 0.22% *w*/*w*), and white rice (8.01 ± 0.18% *w*/*w*) (Table 1). For cordycepin production, a maximum value of 9.85 ± 0.26 mg/g was achieved by supplementation with wheat, followed by Riceberry rice (8.93 ± 0.18 mg/g), while the lowest cordycepin content was equally obtained by supplementation with barley (6.36 ± 0.27 mg/g) and white rice (6.61 ± 0.04 mg/g). In general, cordycepin was widely produced from the fruiting bodies of *C. militaris* using white rice as a solid medium. The cordycepin content produced can vary in the range of 3.57–9.17 mg/g [21,22]. The results presented in our study indicate that defatted rice bran supplemented with different cereal grains—especially wheat—could be employed as an alternative source of cordycepin production from *C. militaris*.

After harvesting the fruiting bodies, the SBRs composed of mycelium and the remaining solid media were collected. They were labeled SBR-B, SBR-R, SBR-Rb, and SBR-W for the residues formed in the presence of barley, white rice, Riceberry rice, and wheat, respectively. The appearance of SBRs dried in a hot-air oven at 45 ± 1 °C for 2 days is depicted in Figure 1e–h. Their solid content was not significantly different, ranging from 26.05 ± 1.83 to 27.24 ± 0.66% (*w*/*w*). As shown in Table 1, the yield of SBRs differed significantly in accordance with the type of medium, whereby the highest yield of 60.54 ± 2.99% (*w*/*w*) was obtained by barley, followed by white rice (54.40 ± 3.75% *w*/*w*), wheat (49.6 6 ± 1.64% *w*/*w*), and Riceberry rice (41.11 ± 0.75% *w*/*w*). Therefore, it can be suggested that from the cultivation of *C. militaris* the yield of SBRs was approximately 3–7 times greater than that of the fruiting bodies. However, the cordycepin content of the SBRs ranged between 0.78 ± 0.14 and 1.09 ± 0.15% (*w*/*w*) and was significantly lower than that found in the fruiting bodies.

Our study is consistent with previous studies that found that cordycepin was found more in the fruiting bodies than in the mycelium of *C. militaris* [7,23]. Therefore, SBRs were further explored for other bioactive components, such as phenolic compounds and polysaccharides; this could be a utilization of waste that remains in large amounts after harvesting the fruiting bodies from *C. militaris* cultivation.

#### 3.1.2. Proximal Compositions of SBRs

Proximate analysis is commonly used for determining moisture and five nutritional compositions including ash, fiber, lipid, protein, and carbohydrate contents. The results of the proximal compositions presented in Table 2 show that all contents were significantly different between the dry SBRs obtained from different culture media (*p* < 0.05). The moisture content of SBRs remained in the range of 9.51 ± 0.03–10.99 ± 0.12% (dw). It can be observed that the SBRs were rich in carbohydrates and protein. Carbohydrates were found to be the main component of SBRs (42.72 ± 0.07–53.00 ± 0.04%dw), followed by crude protein (18.33 ± 0.02–25.67 ± 0.10% dw). In addition, fiber was highly detected in the range of 10.22 ± 0.04–13.08 ± 0.02 % dw. The value of ash content, which identifies inorganic materials such as minerals, ranged from 6.92 ± 0.04 to 9.15 ± 0.05 % dw. In contrast, lipids were found at the lowest level (0.54 ± 0.04–2.00 ± 0.07% dw).

As shown in Table 2, as compared between the different culture media, SBR-R provided the highest carbohydrate content, whereas the highest protein and ash contents were observed in SBR-W (*p* < 0.05). For fiber content, SBR-B evidently resulted in the highest value, while the highest lipid content of 2.00 ± 0.07% dw was present in SBR-Rb. These results indicate that the nutritional compositions of SBRs can be influenced by the solid medium used for the cultivation of *C. militaris*.

The SBRs used in this study consisted of mycelia and the remaining solid media after cultivation. As previously reported, the carbohydrate and crude protein contents in fungal mycelia collected from submerged cultivation were revealed to have wide ranges of approximately 39.6–74.3% and 11–39.5%, respectively [24,25]. These variations could occur based on differences in the strains, ecological origins, or culture conditions [9]. In addition, fibrous polysaccharides such as chitin, cellulose, and glucan are also largely contained in mushroom mycelia. Rice bran and cereal grains are commonly known as rich sources of carbohydrates. For cereal grains, their components have been reported to contain around 71–87% carbohydrates, 6–12% crude protein, and 0.5–3% crude fat [26,27]. Moreover, cereals also have abundant dietary fibers, such as cellulose, lignin, glucans, and water-soluble gums. Consequently, it can be suggested that the proximate compositions of SBRs could depend mainly on the chemical composition of the mycelia obtained from cultivation, as well as the type of solid substrate used.

### 3.2. Extraction Yield of Crude SBR Extracts

The SBRs left over from the cultivation of *C. militaris* with different culture media were used to prepare crude extracts using solid–liquid extraction with different solvents. This extraction method is a one-separation process wherein certain substances are dissolved out from a solid matrix by a liquid solvent, forming a solution. This technique has been preferably used for the extraction of various bioactive compounds from natural sources. In the present study, solid–liquid extraction was performed based on a shaking method, which is more traditional and conventional compared with other extraction methods.

Polarity, solubility, and temperature have been reported to play potential roles in bioactive compound extraction. In this study, various solvents with different polarities were employed to extract bioactive compounds from SBRs—mainly phenolic compounds and polysaccharides. These include ethyl acetate, acetone, ethanol, 50% (*v*/*v*) acetone, 50% (*v*/*v*) ethanol, and DI water. The polarity of the solvents specified in terms of polarity index (PI) can be ordered as follows: water (PI = 9.0) > ethanol (PI = 5.2) > acetone (PI = 5.1) > ethyl acetate (PI = 4.4) [28]. In addition, hot water was also used to examine the thermal effect of this extraction.

The results revealed that the extraction yields ranged from 1.00 ± 0.10 to 24.60 ± 0.20% (*w*/*w*), indicating that the extraction solvent had a significant influence on the yield of the crude extracts (Figure 2). Water offered the highest yields (18.80 ± 0.57–24.60 ± 0.20% *w*/*w*) of crude extract when compared with the other solvents. Unexpectedly, high temperatures for water extraction noticeably decreased the yield of crude extracts. It is possible that the high temperature could degrade some of the compounds represented in the SBRs. Although the heat process can help to increase extraction yield by breaking up the cell wall and accelerating mass transfer, it has been reported that extraction yield can be reduced after the temperature is increased to 55 °C, due to protein denaturation and solubility [29]. In addition, the destruction of heat-sensitive compounds and unstable compounds during the thermal extraction process might lead to a decrease in extraction yield.

Moderate yields, ranging from 10.60 ± 0.32 to 14.40 ± 0.75% (*w*/*w*), were obtained by using the mixed solvents of water and ethanol (50% *v*/*v* ethanol), as well as water and acetone (50% *v*/*v*), while the lowest yields (1.00 ± 0.10–3.00 ± 0.13% *w*/*w*) were present when using a single organic solvent, i.e., ethanol, acetone, and ethyl acetate. These findings emphasize that solvent polarity plays a potential role in extraction, and water can be considered an excellent solvent for providing the maximum yield of crude extract prepared from SBRs. It could be possible that the SBRs contain more water-soluble compounds that could be easily solubilized by water. Similarly, the study by Zhang et al. [30] found that acidified water provided the highest extraction yield (43.52 ± 1.16% *w*/*w*) of crude extract prepared from wheat after fermentation with *C. militaris*, which was higher than the 70% ethanol and 70% acetone extracts.

The crude extract yields were also significantly affected by the type of SBR (Figure 1). As carried out by water extraction, the highest yields of 24.60 ± 0.20 and 24.00 ± 0.63% (*w*/*w*) were obtained by using SBR-B and SBR-W, respectively, followed by SBR-Rb (21.20 ± 0.73% *w*/*w*) and SBR-R (18.80 ± 0.57% *w*/*w*). Moreover, a significant difference in the yields compared between different SBRs was also observed in other solvents. Essentially, different culture media can influence changes in the growth rate and mycelial production of *C. militaris*. Hung et al. [31] cultured *C. militaris* with five different submerged liquid culture media and found that mycelial biomass, as well as extracellular and intracellular polysaccharides, was differently produced through the different culture media types. These phenomena could affect the crude extract yield prepared from *C. militaris*.

The extraction solvent can affect the color of crude SBR extracts. In this study, the crude extract solution (20 mg/mL) was prepared in order to observe the color appearance. As represented in Figure 3, the color shade of crude extracts ranged from yellow to reddish-brown in color. When a low-polarity solvent was used, the crude extracts were clear yellow in color (Figure 3a–c); their intensity seemed to be different when observed optically. The intense brown color of the crude extracts was present when water and aqueous mixtures (50% (*v*/*v*) acetone, and 50% (*v*/*v*) ethanol) were used as extraction solvents (Figure 3d–g). Different colors of crude extracts from SBRs prepared using different solvents might reflect the composition of chemical substances in each crude extract. In general, polyphenols are well known as important secondary metabolites that are widely found in plants. However, they can cause browning via chemical and enzymatic oxidation when exposed to light and heat; this phenomenon can give the product a brown color. In the present study, it is possible that the reddish-brown extracts were rich in polyphenols. The darkened color might have been a result of the oxidation of some polyphenols during dry sample preparation and extraction.

In cosmetics, the color of the ingredients must be considered, so as to prevent disturbing the physical appearance of cosmetic products. Therefore, the amount or concentration of colored ingredients should be considered. In general, natural extracts or active ingredients used in cosmetic preparations are typically added in low quantities—approximately 0.1–1.0% *w*/*w*. In our study, a preparation of 20 mg/mL of crude extract is equivalent to 2% *w*/*v*. This concentration tested might be high enough to confirm that the crude extract from SBRs could be incorporated in cosmetic formulations without disturbing their color.

### 3.3. Total Phenolic, Total Flavonoid, and Total Carbohydrate Contents of Crude SBR Extracts

*C. militaris* is well known to contain many active compounds, including cordycepin, adenosine, polysaccharides, ergosterol, mannitol, and phenolic compounds. These compounds show bioactivities with potential roles in pharmacological and medicinal applications. However, they have rarely been reported in cosmetic applications. Generally, the fruiting bodies and mycelia of *C. militaris* obtained from both the submerged state and solid-state fermentation can be employed as a source for the extraction of bioactive compounds. In our study, the SBRs, which are considered waste after harvesting the fruiting bodies, were obtained to add value by using them as a new alternative source for the extraction of bioactive compounds.

In this study, phenolic compounds were the main focus, and were extracted from the SBRs using different solvents. Phenolics are secondary metabolites that have been broadly studied, characterized, and isolated from several plants; they include phenolic acids, flavonoids, tannins, stilbenes, and lignans [32]. In cosmetics, phenolic compounds are commonly used as active ingredients, with the claim of multifunctional properties such as antioxidant, anti-aging, anti-wrinkle, anti-tyrosinase, anti-pollution, anti-glycation, anti-hyaluronidase, anti-collagenase, anti-elastase, anti-inflammatory, photoprotection, and microbial preservation activities [32,33,34,35]. To date, mushrooms and other microorganisms have been studied as alternative sources of potential phenolic compounds. Therefore, phenolic compounds in SBRs were investigated and evaluated for possible use in cosmetics.

The contents of total phenolic compounds in different SBRs extracted using different solvents are shown in Table 3, demonstrating that the solvent had a significant effect on the extraction of phenolic compounds from the SBRs. The phenolic compounds possess diverse chemical structures, ranging from simple to polymerized forms, and can be distributed in different extraction solvents based on solubility behaviors [36]. High TPC values were achieved by using hot water (46.72 ± 0.46–53.56 ± 0.64 mg GAE/g extract), 50% (*v*/*v*) acetone (41.13 ± 0.89–48.03 ± 1.72 mg GAE/g extract), and 50% (*v*/*v*) ethanol (40.68 ± 0.91–44.08 ± 0.59 mg GAE/g extract). The extraction with water diminished the TPC value, yielding a range of 25.65 ± 0.36–30.86 ± 0.26 mg GAE/g extract. The lowest TPC values were obtained when acetone, ethanol, and ethyl acetate were employed. These findings indicate that the extraction of phenolic compounds from the SBRs of *C. militaris* can be efficiently enhanced by using either high temperatures or 50% (*v*/*v*) mixture solvents of water–ethanol and water–acetone. It is possible that high temperature could assist in the degradation of phenolic compounds that are strongly bound to cell wall components, whereas the mixture solvents might be suitable for extrusion of the free phenolic compounds. These results are similar to those of the study of Sengkhamparn and Phonkerd, in which the total phenolic content in the bitter mushroom (*Tylopilus alboater*) was decreased after the heating process with boiling and steaming [37].

Furthermore, the TPC values demonstrated in Table 3 were also significantly different depending on the type of SBR (*p* < 0.05). When hot water extraction was employed, the highest TPC value of 53.56 ± 0.64 mg GAE/g extract was achieved by SBR-B, followed by SBR-W, SBR-Rb, and SBR-R with values of 49.89 ± 0.55, 47.10 ± 0.86, and 46.72 ± 0.46 mg GAE/g extract, respectively. In contrast, the highest TPC values of 48.03 ± 1.72 and 44.08 ± 0.59 mg GAE/g extract were found in SBR-Rb when 50% acetone and 50% ethanol were employed, respectively. Similar to a previous study by Dong et al. [7], the solid waste medium obtained from the solid-state cultivation of *C. militaris* with unpolished rice was used for the extraction of phenolic compounds. The results showed that the TPC value was equal to 6.03 ± 0.04 mg GAE/g sample when 95% (*v*/*v*) ethanol was used as the solvent. Therefore, it can be emphasized that the solid waste obtained from *C. militaris* cultivation could be used as a source for the extraction of phenolic compounds. However, the types of solid media used for cultivation, along with the solvent used for extraction, are both potential factors that could influence the extraction efficiency.

To further explore phenolic compounds in the crude SBR extracts, total flavonoid content (TFC) was also investigated. Flavonoids are the largest group of phenolic compounds, which are most abundant in plants; they have also been reported in fungi—especially in mushrooms—and endophytes [38,39]. As shown in Table 3, the highest potential values of TFC were gained by using 50% (*v*/*v*) ethanol and 50% (*v*/*v*) acetone which ranged from 18.08 ± 0.14 to 20.93 ± 0.37 and from 14.61 ± 0.26 to 19.58 ± 0.32 mg QE/g extract, respectively, followed by hot water (11.05 ± 0.53–13.59 ± 0.44 mg QE/g extract), and water (7.60 ± 0.03–9.52 ± 0.07 mg QE/g extract). The lowest TFC values were present when using ethanol and acetone, whereas there were no flavonoids detectable in ethyl acetate. These results seemed to contrast with the TPC results mentioned above, in which the use of mixed solvents was more effective for flavonoid extraction than hot water. Kaewnarin et al. [40] reported that methanol and water extracts of wild edible mushrooms contained moderate polar flavonoids, such as flavonoid aglycones, and high polar flavonoids, such as flavonoid glycosides. Therefore, it is possible that the mixed solvent system of the organic solvent with water might have the potential for flavonoid extraction from the SBRs of *C. militaris*. Remarkably, the highest TFC values were gained from SBR-Rb (20.93 ± 0.37 mg QE/g extract) and SBR-W (20.42 ± 0.20 mg QE/g extract), using 50% (*v*/*v*) ethanol as a solvent.

Furthermore, the total carbohydrate content (TCC) in the crude SBR extracts was also examined using the phenol–sulfuric method. This content includes the amounts of simple sugars, oligosaccharides, and polysaccharides represented in the sample tested. The results shown in Table 1 demonstrate that the highest TCC values, ranging between 52.00 ± 0.68 and 59.95 ± 1.03 mg GE/g extract, were detected in the SBRs extracted with hot water. The moderate TCC values were found in the SBRs extracted with 50% (*v*/*v*) ethanol, 50% (*v*/*v*) acetone, and water, which provided values in the ranges of 29.19 ± 1.63–41.88 ± 1.14, 26.93 ± 1.60–30.85 ± 1.09, and 18.51 ± 0.94–25.51 ± 0.54 mg GE/g extract, respectively. The lowest TCC value was observed in the presence of ethanol, while ethyl acetate and acetone were unable to detect the TCC.

Many researchers have focused on and had an interest in the production of polysaccharides from *C. militaris* due to their biological activities and health benefits [31,41,42]. As studied by Yan et al. [43], galactomannan is a main-type polysaccharide found in *C. militaris*. The polysaccharides found in *C. militaris* can be either extra- or intracellular polysaccharides. The extracellular polysaccharides can be easily secreted into the culture filtrate via submerged fermentation [44]. For intracellular polysaccharides, they were mostly extracted from the mycelial part, and solid substrates were fermented with *C. militaris* using hot or boiling water [31,45]. The TCC shown in Table 3 can be extensively found in various solvents, except for ethyl acetate and acetone. Thus, it can be hypothesized that not only might polysaccharides be found in the SBR extracts, some oligosaccharides and simple sugars (mono- and disaccharides) might also be present. They could be possibly produced due to the enzymatic degradation of the polysaccharides represented in the solid substrate, mainly composed of starch and cellulose. The two main enzymes involved are amylase and cellulase, which can be secreted from fungi during SSF with a carbohydrate-rich substrate [46]. Essentially, simple sugars and oligosaccharides can be extracted with water and polar solvents due to their multiple OH groups contained in the molecule structure. As reported by Suna et al. [47] and Lane et al. [48], oligosaccharides extracted by 10% (*v*/*v*) ethanol were more effective than using distilled water alone. In our study, it can be suggested that hot water extraction was more effective for the extraction of carbohydrates from *C. militaris* SBRs. Among the SBRs extracted with hot water, the greatest TCC value (59.95 ± 1.03 mg GE/g extract) was present in SBR-W, followed by SBR-B, SBR-R, and SBR-Rb, with values of 58.17 ± 0.21, 53.31 ± 0.82, and 52.00 ± 0.68 mg GE/g extract, respectively.

In cosmetics, polysaccharides derived from natural sources—especially plants and microorganisms—have been extensively used as functional cosmetic ingredients based on their moisturizing, conditioning, and rheology-modifying properties. In addition, some polysaccharides can also exhibit biological activities that enable them to be used as active ingredients in cosmetic formulations. Recently, alternative natural sources have been extensively explored, owing to growing natural and organic cosmetic trends. Mushroom polysaccharides have become of interest for cosmetic and cosmeceutical applications, due to their beneficial effects. Simple sugars and oligosaccharides can provide hygroscopic effects because their molecules contain polyhydroxyl groups—called polyols—that can exert hydrating and moisturizing effects on skin and hair. Interestingly, oligosaccharides have also served as prebiotics, which are currently being used as active ingredients in microbiome skincare formulations [49,50]. Therefore, it can be summarized that the SBRs from *C. militaris* could represent an interesting new alternative source of bioactive compounds for developing new ingredients in cosmetics and cosmeceuticals.

### 3.4. Antioxidant Activities of Crude SBR Extracts

Free radicals such as reactive oxygen species (ROS) and reactive nitrogen species (RNS) generated from cellular metabolisms and various environmental conditions—mainly UV rays, air pollutants, chemicals, and industrial waste—are considered to be unstable molecules that can damage DNA, proteins, and lipids, leading to diseases and health problems. Moreover, they can damage skin cells, which can result in skin aging, wrinkles, skin dullness, and crow’s feet, along with dark spots, dark circles under the eyes, skin dryness, and loss of skin elasticity. To overcome these phenomena, antioxidants are employed to act against free radicals via different mechanisms.

Antioxidants have long played a role as important ingredients in cosmetic preparations. In cosmetic formulations, they can serve as (1) additive ingredients (stabilizer) used for protecting and stabilizing other cosmetic ingredients against oxidative effects, or (2) active ingredients developed from in vitro and in vivo research. They are commercially used in cosmetics, as presented in the form of both antioxidant compounds and crude extracts, which are endorsed for many purposes, such as anti-aging, anti-wrinkle, and anti-pollution effects, as well as boosting skin radiance. Furthermore, they can be either chemically synthesized or naturally derived ingredients. Nevertheless, natural antioxidants tend to be preferred due to the side effects of synthetic antioxidants, along with the arrival of natural trends in cosmetics. Natural antioxidants comprise a group of phenolic compounds, polysaccharides, proteins, peptides, vitamins, carotenoids, and minerals that are be available in natural resources; among them, plant polyphenols and phenolic acids have been the most studied and focused on for cosmetic applications [32,33,51]. At present, new alternative sources—especially fungi—are gaining interest in the study and investigation of their constituents for further development as natural antioxidants.

The antioxidant capacity of crude SBR extracts was proven and confirmed by investigating antioxidant activities based on DPPH radical scavenging activity, ABTS radical scavenging activity, and FRAP assays. The results depicted in Figure 4 show that the antioxidant activities were significantly influenced by the type of SBR and the type of solvent (*p* < 0.05). The DPPH assay is considered an excellent tool for determining the free radical scavenging activity of samples based on their hydrogen-donating action. As depicted in Figure 4a, hot water extraction provided the optimal values of DPPH radical scavenging activity in all crude SBR extracts, ranging from 40.12 ± 0.85 to 66.62 ± 2.10 mg TEAC/g extract. It was found that the use of hot water gave much higher values of antioxidant activity—approximately 3–5-fold—than the values obtained from unheated water extraction (12.53 ± 0.08–14.41 ± 0.33 mg TEAC/g extract). The mixed-solvent systems (50% (*v*/*v*) acetone and 50% (*v*/*v*) ethanol) seemed to provide similarly moderate values, ranging from 21.83 ± 0.44 to 26.42 ± 0.32 and from 20.06 ± 0.48 to 23.350 ± 0.446 mg TEAC/g extract, respectively, whereas low values of antioxidant activity were present in ethyl acetate, acetone, and ethanol. Therefore, it can be said that hot water can optimize the DPPH radical scavenging activity of the SBR extracts from *C. militaris*. Among the hot water extracts, SBR-B exhibited the highest values of DPPH radical scavenging activity (66.62 ± 2.10 mg TEAC/g extract), followed by SBR-W, SBR-Rb, and SBR-R, with values of 54.81 ± 0.37, 51.05 ± 0.63, and 40.12 ± 0.85 mg TEAC/g extract, respectively (*p* < 0.05).

The ABTS assay investigates the antioxidant capacity to scavenge ABTS free radicals. As shown in Figure 4b, low values of antioxidant activity were still found in the presence of organic solvents. Remarkably, it was observed that high values of ABTS antioxidant activity could be obtained by using heated and unheated water, as well as water mixtures. Nevertheless, differences in the activities could be observed between the types of SBR. As compared with the results within the same solvent, SBR-B provided the highest antioxidant activity against ABTS radicals, with 212.00 ± 3.43 and 168.17 ± 2.42 mg TEAC/g extract when using hot water and 50% (*v*/*v*) ethanol as solvents, respectively. Meanwhile, SBR-Rb and SBR-W showed the highest values when extracted with either water (161.08 ± 2.85 and 164.74 ± 3.18 mg TEAC/g extract, respectively) or 50% (*v*/*v*) acetone (168.15 ± 4.44 and 170.11 ± 0.83 mg TEAC/g extract, respectively).

The antioxidant activity of the crude SBR extracts assayed by FRAP showed a similar trend to the results of the DPPH assay (Figure 4c). It can be seen that the ferric-reducing ability of the crude extracts tended to depend on the solvent, in the following order: hot water > 50%(*v*/*v*) acetone = 50%(*v*/*v*) ethanol > water > acetone = ethanol > ethyl acetate. As observed in the hot water extracts, SBR-B showed the highest FRAP value (101.62 ± 4.42 mg TEAC/g extract), followed by SBR-W (94.48 ± 3.289 mg TEAC/g extract), while there was no significant difference in FRAP values between SBR-R and SBR-Rb, which were equal to 87.35 ± 2.07 and 85.22 ± 1.77 mg TEAC/g extract, respectively (*p* ≥ 0.05).

Therefore it can be summarized that the antioxidant activities of the crude SBR extracts can be optimized and improved by using hot water extraction. Indeed, water is low in price, quite safe, and legally permitted to be used in the food and cosmetic industries. In addition, it is also eco-friendly, and considered to be the best solvent for green and sustainable extraction methods. Among the SBRs extracted using hot water, SBR-B seemed to exhibit the best potential antioxidant activities, followed by SBR-W, SBR-Rb, and SBR-R. Remarkably, these findings support previous studies that found that antioxidant capacity can be gained not only from the whole parts of *C. militaris*—i.e., fruiting bodies and mycelia [7,31,52]—but also from solid wastes or residues left from its cultivation [53]. As demonstrated by Rupa et al. [54], *C. militaris* extract encapsulated in nanoemulsion exhibited significantly higher antioxidant activity than the free extract. This could be a promising approach for cosmetic and cosmeceutical applications.

Today, there is a growing trend in the use of natural antioxidants as active ingredients in anti-aging products for facial and body skin. Likewise, the antioxidant-activity- and antiwrinkle-activity-based anti-aging potential in the fruiting bodies and mycelial parts of several mushrooms has been the subject of greater focus and interest. Many types of edible mushrooms—such as *Agaricus bisporus*, *Auricularia* spp., *Ganoderma lucidum*, *Grifola frondosa*, *Lentinula edodes*, *Pleurotus eryngii*, *Pleurotus ostreatus*, and *Tremella fuciformis*—have been proven to potentially display cosmetic and cosmeceutical properties (mainly antioxidant capacity) [40,55,56]. Moreover, mushrooms from the Cordyceps family—including *Ophiocordyceps sinensis* (syn. *C. sinensis*) and *C. cicadae*—have also exhibited antioxidant and anti-aging activities [57,58]. *C. militaris* could be an interesting source for the preparation of natural antioxidants to be used as anti-aging ingredients. In addition, the SBRs left over from the cultivation can be also used as alternative sources to the fruiting bodies and the mycelia.

### 3.5. Tyrosinase Inhibitory Activity of Crude SBR Extracts

Melanogenesis is a complicated process that involves both enzymatic and chemically mediated processes for skin pigmentation. The exogenous causes—especially UV radiation—mainly increase melanin synthesis and accumulation, leading to skin problems such as melasma, age spots, freckles, and the risk of skin cancer. Tyrosinase—a copper-containing enzyme in the melanosome—is the key enzyme involved in melanin synthesis via the oxidation of L-tyrosine to dopaquinone. For this reason, an attempt to inhibit tyrosinase is a focused target for the development of skin-whitening agents as active ingredients in cosmetics and cosmeceuticals, because of their high specificity and safe and low side effects [59].

In this study, the anti-tyrosinase activity of different SBRs extracted using different solvents was investigated. The results revealed that all SBR extracts possessed anti-tyrosinase activity, which clearly exerted a polarity-dependent effect (Table 4). The 50% (*v*/*v*) ethanol extract strongly inhibited the tyrosinase enzyme when compared with the other extracts prepared from different solvents. Among them, SBR-Rb exhibited the highest value of 51.13 ± 1.11 mg KAE/g extract, while SBR-B (49.22 ± 0.18 mg KAE/g extract) and SBR-W (48.77 ± 0.79 mg KAE/g extract) showed similar results (*p* ≥ 0.05). The lowest value (42.84 ± 0.66 mg KAE/g extract) was observed in SBR-R. Likewise, the percentage of tyrosinase inhibition of the 50% (*v*/*v*)ethanol extract considerably ranged from 75.76 ± 1.18 to 90.43 ± 1.96% at the crude extract concentration of 1 mg/mL.

When comparing the 50% (*v*/*v*) ethanol extract with the other solvent extracts, the results of 50% (*v*/*v*) acetone and hot water extracts were approximately halved, while the SBR extracted from water and other organic solvents had significantly lower tyrosinase-inhibitory power. Conversely, the study of Chien et al. [60] reported that the water extract prepared from *C. militaris* mycelium using submerged fermentation showed the highest tyrosinase inhibition (around 25%) when compared with the 75% ethanol and 50% ethanol extracts at the same extract concentration (1 mg/mL).

As studied previously, tyrosinase inhibition was initially focused on cordycepin extract prepared from the fruiting bodies of *C. militaris*. Cha and Kim demonstrated that cordycepin can suppress cellular tyrosinase activity and melanin production in cultured B16F0 melanoma cells [61]. Subsequently, Li et al. [62] reported that both cordycepin and the mycelia of *C. militaris* extracted using boiling water can provide tyrosinase-inhibition activity. In our study, the SBRs prepared from *C. militaris*—which mainly consisted of phenolic compounds, flavonoids, and carbohydrates (as illustrated previously in Table 3)—were also considered as potential alternative sources of anti-tyrosinase-activity-based whitening agents. Indeed, phenolic compounds have been reported to exhibit anti-tyrosinase activity [63]. In cosmetics, whitening products mainly help to reduce melanin, lighten skin, and remedy skin issues such as age spots, acne scars, or hormonal discoloration. Natural whitening ingredients are often more attractive and trendy to customers; therefore, the SBR extract—a nature-derived material—could be more interesting for further use as an active ingredient in skin whitening and lightening products.

### 3.6. Correlation Analysis

Several reports have mentioned that the bioactivities of plant or microbial extracts positively corresponded with their compositions [64,65]. To investigate whether there were any relationships between the content of bioactive compounds—i.e., TPC, TFC, and TCC—in the crude SBR extracts versus their biological activities—i.e., DPPH radical scavenging activity, ABTS radical scavenging activity, FRAP, and tyrosinase-inhibitory activity—correlational analysis using Pearson’s correlation coefficient (r) was therefore explored. According to Mukaka [66], an r-value ranging from 0.0–0.3 represents a negligible correlation, 0.3–0.5 represents a weak correlation, 0.5–0.7 represents a moderate correlation, 0.7–0.9 represents a strong correlation, and 0.9–1.0 represents a very strong correlation.

The results depicted in Table 5 show all significant correlations between bioactive compounds (TPC, TFC, and TCC) and antioxidant activities, at a confidence level of 99% (*p* < 0.01). The TPC was very strongly correlated with DPPH (r = 0.833), ABTS (r = 0.912), and FRAP (r = 0.980). These results are similar to those of previous studies, in that the antioxidant activities of the fermented extracts prepared from SSF were strongly related to the TPC values [67,68]. For anti-tyrosinase activity (TYR), its correlation with TPC was observed at a strong level (r = 0.735), while a very strong correlation was found between TFC and TYR (r = 0.864). This suggests that flavonoids could play a more crucial role as anti-tyrosinase agents than the total phenolic compounds found in the SBR extracts. As observed in Table 5, TFC had a very strong correlation with ABTS (r = 0.813) and FRAP (r = 0.804), but was weakly associated with DPPH (r = 0.496). This phenomenon could suggest that DPPH radical scavenging activity in the SBR extracts was not only a function of flavonoids, but that other types of phenolic compounds—such as phenolic acids, tannins, lignans, and stilbenes—might also potentially act as antioxidants in the extract. This is similar to the findings of Alimpić et al. [69], who revealed that IC_50_ values of DPPH radical scavenging activity were more attributed to the total phenolic content (r = -0.74) than the flavonoid content (r = 0.24) in *Salvia amplexicaulis* extract.

Moreover, the correlation between the total carbohydrate content (TCC) and the antioxidant and anti-tyrosinase activities in the crude SBR extracts was also investigated. The results shown in Table 5 indicate that TCC was very strongly correlated with all three antioxidant activities—DPPH (r = 0.885), ABTS (r = 0.852), and FRAP (r = 0.957)—while a weak correlation (r = 0.424) was observed between TCC and anti-tyrosinase activity (TYR). As reported previously, polysaccharides and mannitol are carbohydrates that have been found to be important constituents in *C. militaris.* It is possible that they are ingredients in the SBR extracts of *C. militaris* that express antioxidant activities. This evidence is consistent with previous studies that showed that polysaccharides composed of more mannose and galactose residues can exhibit great antioxidant and anti-glycation activities. In addition, highly positive correlations (r > 0.9) between mannose and antioxidant activities (DPPH, ABTS, and FRAP) have also been proven [70]. Similarly, in the study of Siu et al. [71], the crude polysaccharide isolated from edible/medicinal mushrooms displayed a very strong correlation with antioxidant activities assayed by ABTS and FRAP. Therefore, it can be summarized that antioxidant activities in the SBR extracts of *C. militaris* could be governed by both total phenolics and carbohydrates in the extracts, while the effective tyrosinase-inhibitory activity could be mainly attributed to the higher flavonoid content in the extracts.

### 3.7. Photoprotective Activity of Crude SBR Extracts

UV radiation induces reactive oxygen species (ROS), melanogenesis, and DNA damage, causing skin cell damage, skin erythema (sunburn), pigmentation (tanning), photoaging, and skin cancer. UVB has a short wavelength, from 290 to 320 nm, and provides a high-energy source that directly interacts with skin cells by penetrating the epidermis, leading to critical skin and DNA damage. Meanwhile, UVA has a long wavelength (320–400 nm) that can penetrate the deep skin layers, leading to tanning and photoaging effects. There are many organic and inorganic compounds generally used in sunscreen products that function and are claimed as sunscreen agents, UV filters, and sunscreen boosters. Unfortunately, some of them can be considered allergenic and photosensitizing. Therefore, naturally based sunscreen agents have been more sought after and focused on in modern sunscreen formulations.

It has been reported that many natural compounds containing molecules or molecular complexes can absorb UV radiation to reduce oxidative stress and protect skin cells [72]. Phenolic compounds such as flavonoids and tannic acid have been stated to be able to absorb UV radiation [73,74]. Moreover, some studies have reported that polysaccharides have potential photoprotective capacity against UV rays [75]. In the present study, the UV absorption of crude SBR extracts was detected via scanning on a UV–Vis spectrophotometer at 290 to 400 nm. A concentration of 0.5 mg/mL of all crude extracts presented the ability to absorb UV radiation—mostly effective in the UVA range (Figure 5). The absorption profiles were noticeably influenced in accordance with the types of solvent and SBR, where the solvent seemed to have a greater impact. As shown in Figure 5, all hot water extracts were found to be the most effective UV absorbers—especially in the range of 290–330 nm—when compared to the other extracts. This could be a result of the high contents of both total phenolic compounds and crude polysaccharides present in the hot water extracts. As also reported by Wong et al. [76], the hot water extract from *Cordyceps* mycelia can inhibit UVB-induced oxidative and DNA damage in human fibroblast cells. Therefore, the hot water extract from the SBR of *C. militaris* might be further developed as an active ingredient for enhancing or boosting the efficacy of sunscreen products.

### 3.8. Cell Viability and Proliferation Capability of Crude SBR Extracts

Based on the evidence of bioactive compounds and bioactivities represented in the crude SBR extracts previously mentioned, the 50% (*v*/*v*) ethanol and hot water extracts of the *Cordyceps* SBRs were selected in order to measure their cytotoxicity using fibroblast cell lines (NIH/3T3) treated for 48 h at various concentrations of extract (0.001–10 mg/mL). The 50% (*v*/*v*) ethanol extracts contained high amounts of flavonoids with the highest anti-tyrosinase activity, while the hot water extracts were rich in total phenolic compounds and carbohydrates, as well as potential antioxidant and photoprotective activities. The results of their cytotoxic and proliferative effects are illustrated in Figure 6a,b, respectively. It can be clearly observed that the cell viability was more than 50% after treatment, even at higher doses (10 mg/mL), except for the SBR-B hot water extract tested at 10 mg/mL (46.3%). This indicates the low toxicity of the crude SBR extracts, ensuring their safety. Interestingly, the extracts were able to promote fibroblast proliferation if the cell viability value was significantly higher than that of the control (>100%). It can be observed that the number of available cells was mostly increased in the concentration range of 0.001–0.1 mg/mL, where some extracts promoted proliferation to nearly 200% (twofold cell proliferation). As previously reported, antioxidants have been found to increase the replicative lifespan of fibroblasts—either directly or indirectly—by reduction in oxidative damage, activation of signaling pathways, activation of proteasomes, or hormetic actions [77,78,79]. Hence, the optimal concentration range and bioactive components of the SBR extract could affect fibroblast proliferation capacity.

Fibroblasts play an important role in the production of the extracellular matrix in the dermis layer, including collagen, elastin, fibrin, and hyaluronic acid, which provide firmness and elasticity to the skin. The proliferative and metabolic activity of fibroblasts declines due to the aging process and extrinsic environmental factors, resulting in functional impairment and decreased production of structural substances, leading to signs of skin aging such as wrinkles, loss of elasticity, and skin fragility [77]. The cytotoxicity results of the SBR extracts emphasized their safety for use in cosmetic applications. Remarkably, the ability to promote fibroblast proliferation can contribute as a promising ingredient that could possibly enhance the production of collagen, improve the elasticity of the skin, and restore aging skin in anti-aging and skin-rejuvenating products.

### 3.9. LC–MS/MS Analysis

In this study, identification and quantification of phenolic compounds in the 50% (*v*/*v*) ethanol and hot water extracts of SBRs were performed using LC-QqQ-MS/MS analysis. The phenolic profiles represented in Table 6 reveal that seven phenolic acids (gallic acid, protocatechuic acid, chlorogenic acid, caffeic acid, *p*-coumaric acid, *o*-coumaric acid, and *p*-hydroxybenzoic acid) and four flavonoids (rutin, quercetin, apigenin, and naringenin) were qualified and considered as phenolic markers in the SBR extracts. To the best of our knowledge, these findings were first reported in the present study involving phenolic compounds in the SBRs of *C. militaris*. However, the presence and content of phenolic compounds clearly varied in accordance with the SBR type, which could affect their biological activities. Phenolic acid composition appeared to be similar between the 50% (*v*/*v*) ethanolic and hot water extracts, but the content of phenolic acids in hot water extracts was rather high; in contrast to flavonoid profiles, they were more predominant in the 50% (*v*/*v*) ethanol extracts. 

As shown in Table 6, *p*-coumaric acid and naringenin were the main phenolic acid and flavonoid, respectively, and were found at high levels in all SBR extracts. Interestingly, rutin was also found in both the 50% (*v*/*v*) ethanol and hot water extracts of SBR-Rb and SBR-W, while quercetin was only detected in the 50% ethanol extracts. The highest rutin and quercetin contents were found in the ERb extract which, equal to 151.96 ± 10.07 and 71.67 ± 4.72 µg/g of extract, respectively. It can be observed that most of the phenolic compounds found in the SBR extracts of *C. militaris*—including *p*-coumaric acid, protocatechuic acid, *p*-hydroxybenzoic acid, gallic acid, naringenin, and rutin—have also been reported in the crude extracts of various types of mushroom [40,80,81]. However, their proportions appeared to be significantly different between them (*p* < 0.05).

Other potentially bioactive compounds represented in the 50% ethanol and hot water extracts of SBRs were also screened for the identification and characterization of individual components using LC-QTOF-MS/MS analysis. Table 7 shows the MS data of compounds that were tentatively identified in the SBR extracts. A total of 52 major compounds were obtained and classified into 9 categories: nucleosides, nucleobases, amino acids, peptides, sugars, phospholipids, alkaloids, organic acids, and vitamins. It is possible that these compounds could play a role in the biological activities of the SBR extracts. These findings also support previous studies in which *C. militaris* was rich in adenosine, cordycepin, and various amino acids—especially lysine, glutamic acid, and proline [24,82]. These were detected in all SBR extracts (Table 7). Two sweeteners—trehalose and mannitol—were found in all of the SBR extracts, because they are predominant compounds found in the mycelia of *C. militaris*, as has been previously reported [24,53]. In cosmetics, amino acids and sugars have been generally used and proclaimed as skin-moisturizing, -hydrating, and -toning agents due to their hygroscopic properties. Remarkably, several dipeptides and tripeptides were disclosed, which varied with the type of SBR extract. These could be interestingly used as bioactive peptides for cosmetic applications. Peptides can potentially display anti-aging properties, and have become popular active ingredients in skin anti-aging and rejuvenating skincare products [83].

## 4. Conclusions

The present study highlights that solid-based residues (SBRs) left over from the cultivation of *C. militaris* can potentially be used as a new material source for the preparation of crude extracts for use in cosmetic applications. The bioactive compounds—mainly phenolic compounds and carbohydrates contained in the crude SBR extracts— exhibited multiple functional properties that are sought after in cosmetics, including antioxidant and anti-tyrosinase activities, photoprotection, and cell-proliferative effects, along with low cytotoxicity to fibroblast cell lines. When comparing crude SBR extracts prepared from different types of SBR and extraction solvents, SBR-B extracted using hot water, which had the highest phenolic content, showed excellent antioxidant capacity, and could thus be utilized as an active ingredient in anti-aging and anti-wrinkle products. The highest content of polysaccharides and sugars in SBR-W extracted by hot water also indicates its potential antioxidant capacity, while SBR-Rb extracted by 50% (*v*/*v*) ethanol, which was rich in flavonoids, showed the highest tyrosinase-inhibition activity, and could thus function as an active ingredient in skin-whitening products. Phenolic profiling carried out via the LC–ESI-QqQ-MS/MS technique revealed that seven phenolic acids and four flavonoids were identified in the crude SBR extracts. The hot water extracts contained high amounts of phenolic acids, while the 50% (*v*/*v*) ethanol extracts were rich in flavonoid content. *p*-Coumaric acid and naringenin can be used as the phenolic markers of the crude SBR extracts. Additionally, rutin and quercetin were detected in SBR-Rb and SBR-W. LC-QTOF-MS/MS analysis indicated that other compounds—including nucleosides, nucleobases, amino acids, peptides, sugars, phospholipids, alkaloids, organic acids, and vitamins—were also found in the crude SBR extracts. SBRs are therefore promising materials for the further development of new active ingredients in cosmetics and related fields. Nevertheless, further assessments of their other biological activities and stability, along with clinical studies, are necessary in order to develop and ensure the use of these crude extracts as multifunctional ingredients in cosmetic and cosmeceutical formulations.

## Figures and Tables

**Figure 1 jof-07-00973-f001:**
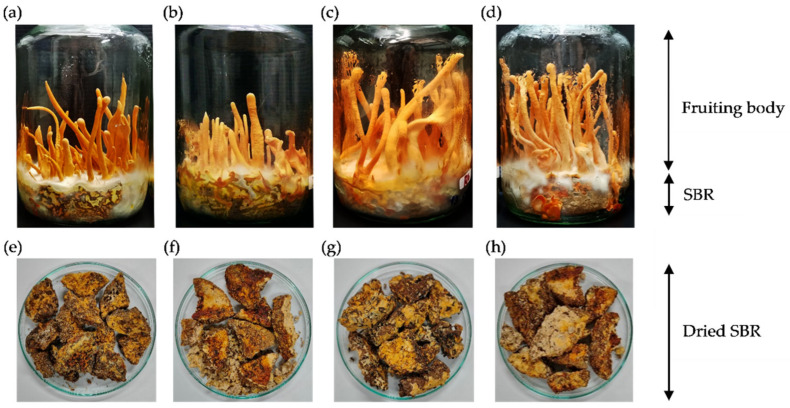
Morphology of *C. militaris* cultivated on solid media containing defatted rice bran and different types of cereals, and the appearance of dry solid-based residues (SBRs) prepared from different culture media: (**a**,**e**) barley, (**b**,**f**) white rice, (**c**,**g**) Riceberry rice, and (**d**,**h**) wheat.

**Figure 2 jof-07-00973-f002:**
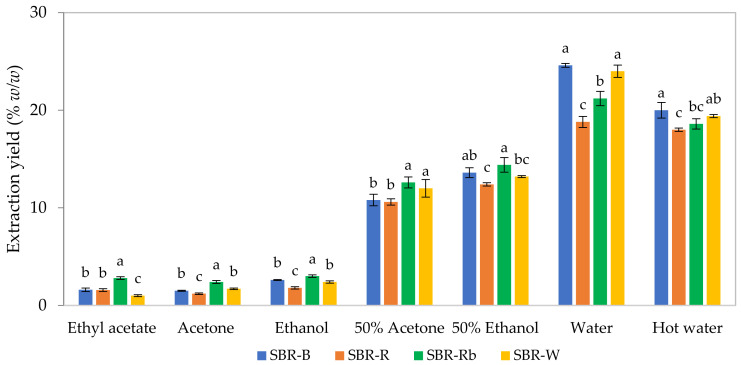
Extraction yield of crude extracts from solid-based residues (SBRs) prepared using different solvents. Different superscript letters compared within the same solvent are significantly different based on Tukey’s test (*p* < 0.05).

**Figure 3 jof-07-00973-f003:**
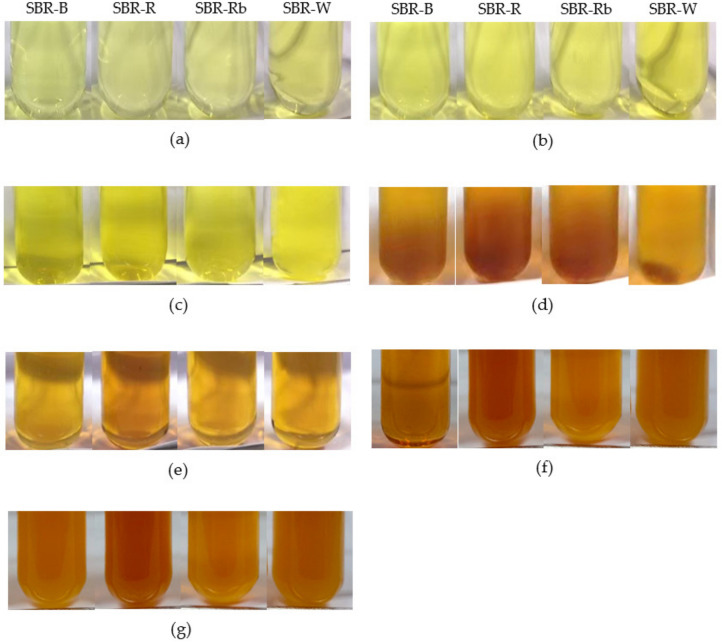
Color of crude extract solutions (20 mg/mL) prepared from different solid-based residues (SBRs) with different solvents: (**a**) ethyl acetate; (**b**) acetone; (**c**) ethanol; (**d**) 50% (*v*/*v*) acetone; (**e**) 50% (*v*/*v*) ethanol; (**f**) water; and (**g**) hot water.

**Figure 4 jof-07-00973-f004:**
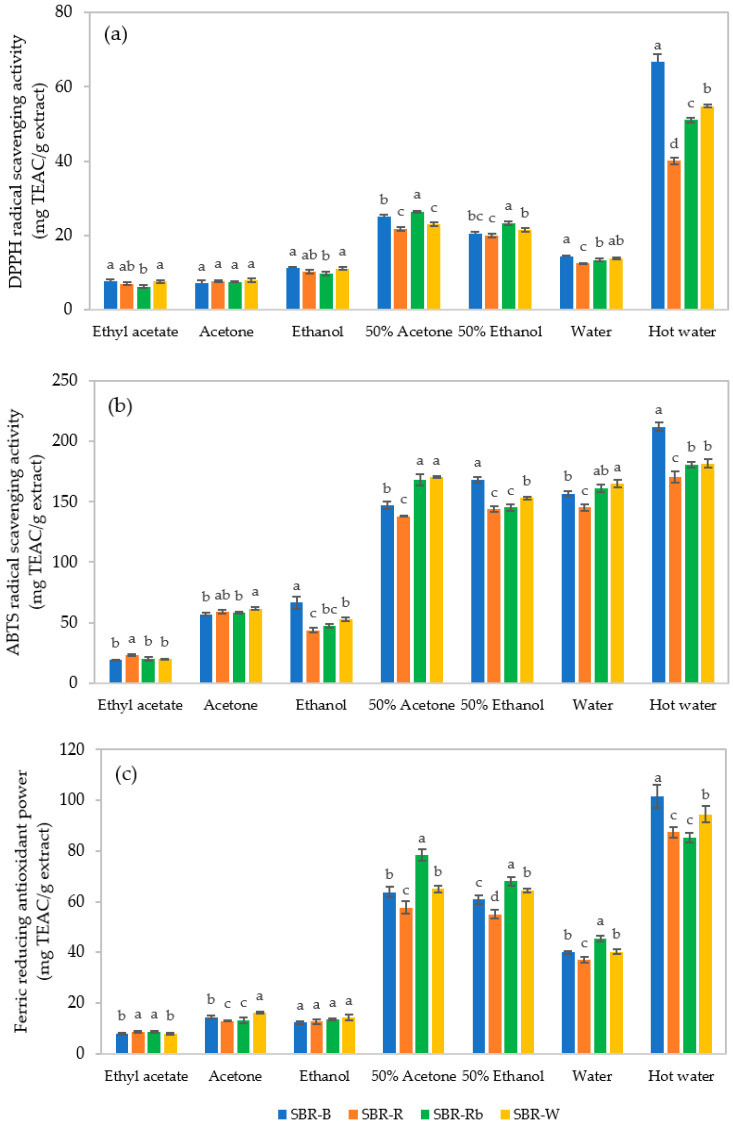
Antioxidant activities of crude extracts prepared from solid-based residues (SBRs) using different solvents: (**a**) DPPH radical scavenging activity; (**b**) ABTS radical scavenging activity; (**c**) ferric-reducing antioxidant power (FRAP). Values compared for the same solvent with different superscript letters are significantly different based on Tukey’s test (*p* < 0.05).

**Figure 5 jof-07-00973-f005:**
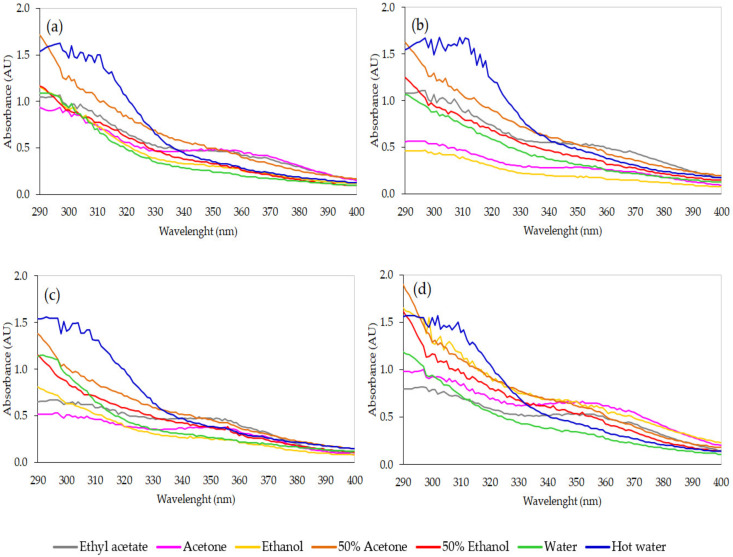
UVB and UVA absorption patterns of crude extracts prepared from solid-based residues (SBRs) using different solvents: (**a**) SBR-B; (**b**) SBR-R; (**c**) SBR-Rb; and (**d**) SBR-W.

**Figure 6 jof-07-00973-f006:**
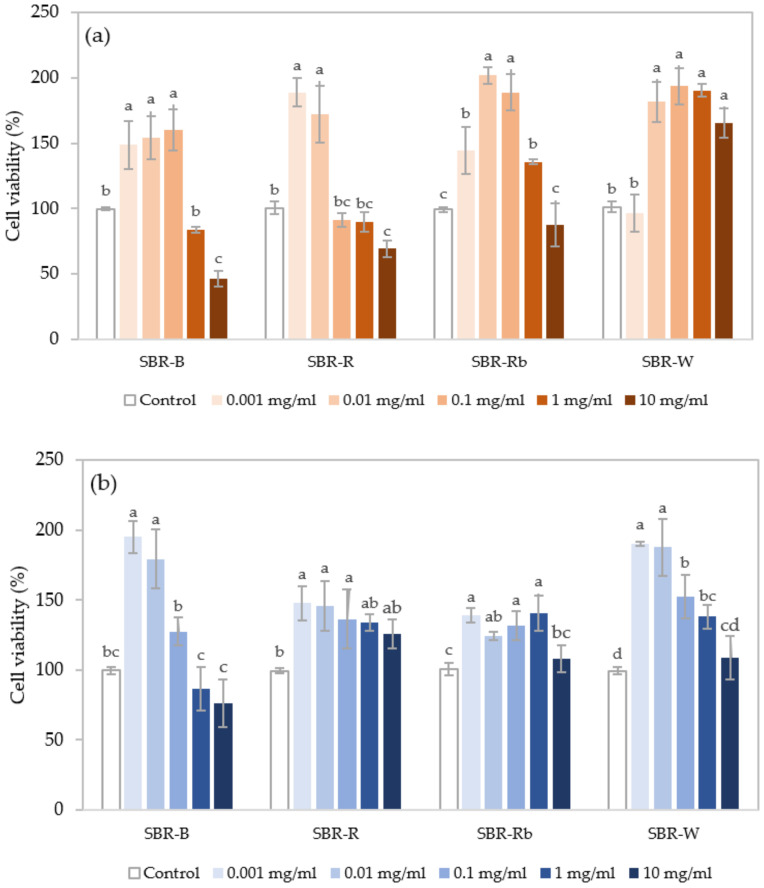
Cell viability effects of (**a**) 50% (*v*/*v*) ethanolic extracts and (**b**) hot water extracts prepared from different solid-based residues (SBRs) of *C. militaris* on fibroblast cell lines (NIH/3T3) at different concentrations (0.001–10 mg/mL). Values compared for the same SBR with different superscript letters are significantly different based on Tukey’s test (*p* < 0.05).

**Table 1 jof-07-00973-t001:** Yields of fruiting bodies and solid-based residues (SBRs), and cordycepin content obtained from the cultivation of *C. militaris* with different culture media.

Culture Media	Yield of Dry Sample (% *w*/*w*) *	Cordycepin Content in Dry Sample (mg/g)
*Fruiting body*		
Rice bran + Barley (B)	8.91 ± 0.22 ^c^	6.36 ± 0.27 ^c^
Rice bran + White rice (R)	8.01 ± 0.18 ^d^	6.61 ± 0.04 ^c^
Rice bran + Riceberry rice (Rb)	12.85 ± 0.37 ^a^	8.93 ± 0.18 ^b^
Rice bran + Wheat (W)	11.61 ± 0.32 ^b^	9.85 ± 0.26 ^a^
*SBR*		
Rice bran + Barley (B)	60.54 ± 2.99 ^a^	0.78 ± 0.14 ^a^
Rice bran + White rice (R)	54.40 ± 3.75 ^b^	0.97 ± 0.08 ^a^
Rice bran + Riceberry rice (Rb)	41.11 ± 0.75 ^c^	1.09 ± 0.15 ^a^
Rice bran + Wheat (W)	49.66± 1.64 ^b^	1.06 ± 0.11 ^a^

Values are expressed as the mean ± standard deviation. Means compared among the same part (fruiting body or SBR) in the same column with different superscript letters are significantly different based on Tukey’s test (*p* < 0.05); *: g of dry sample/100 g solid substrate.

**Table 2 jof-07-00973-t002:** Proximate analysis of dried solid-based residues (SBRs) obtained from the cultivation of *C. militaris* with different types of solid media.

Parameter (% dw) *	Dried Waste Material **			
	SBR-B	SBR-R	SBR-Rb	SBR-W
Moisture	10.12 ± 0.06 ^b^	10.99 ± 0.12 ^a^	9.86 ± 0.04 ^c^	9.51 ± 0.03 ^d^
Ash	7.35 ± 0.08 ^c^	6.92 ± 0.04 ^d^	8.75 ± 0.04 ^b^	9.15 ± 0.05 ^a^
Fiber	13.08 ± 0.02 ^a^	10.22 ± 0.04 ^d^	12.72 ± 0.05 ^b^	12.19 ± 0.02 ^c^
Lipid	0.72 ± 0.03 ^b^	0.54 ± 0.04 ^c^	2.00 ± 0.07 ^a^	0.76 ± 0.05 ^b^
Protein	21.24 ± 0.47 ^b^	18.33 ± 0.02 ^c^	21.80 ± 0.68 ^b^	25.67 ± 0.10 ^a^
Carbohydrate	47.50 ± 0.05 ^b^	53.00 ± 0.04 ^a^	44.88 ± 0.08 ^c^	42.72 ± 0.07 ^d^

Values are expressed as the mean ± standard deviation. Means in the same column with different superscript letters are significantly different based on Tukey’s test (*p* < 0.05). *: dw (dry weight); **: SBR obtained from solid media containing barley (B), white rice (R), Riceberry rice (Rb), and wheat (W).

**Table 3 jof-07-00973-t003:** Total phenolic content (TPC), total flavonoid content (TFC), and total carbohydrate content (TCC) of solid-based residues (SBRs) extracted using different solvents.

Conditions		TPC (mg GAE/g Extract)	TFC (mg QE/g Extract)	TCC (mg GE/g Extract)
Solvent	Residue			
Ethyl acetate	SBR-B	13.99 ± 1.26 ^b^	n.d.	n.d.
	SBR-R	14.16 ± 0.78 ^b^	n.d.	n.d.
	SBR-Rb	16.54 ± 1.16 ^a^	n.d.	n.d.
	SBR-W	15.41 ± 0.68 ^ab^	n.d.	n.d.
Acetone	SBR-B	19.06 ± 0.81 ^a^	0.89 ± 0.14 ^a^	n.d.
	SBR-R	14.84 ± 1.18 ^b^	0.74 ± 0.08 ^a^	n.d.
	SBR-Rb	16.20 ± 0.30 ^b^	0.71 ± 0.06 ^a^	n.d.
	SBR-W	19.81 ± 0.80 ^a^	0.68 ± 0.03 ^a^	n.d.
Ethanol	SBR-B	18.20 ± 0.59 ^c^	1.84 ± 0.21 ^a^	4.39 ± 0.91 ^ab^
	SBR-R	19.79 ± 0.63 ^b^	1.06 ± 0.13 ^b^	3.32 ± 0.37 ^b^
	SBR-Rb	15.24 ± 0.20 ^d^	2.25 ± 0.30 ^a^	3.62 ± 0.20 ^b^
	SBR-W	21.53 ± 0.21 ^a^	1.90 ± 0.32 ^a^	5.16 ± 0.47 ^a^
50% Acetone	SBR-B	44.04 ± 1.44 ^b^	16.98 ± 0.59 ^c^	30.85 ± 1.09 ^a^
	SBR-R	41.13 ± 0.89 ^c^	14.61 ± 0.26 ^d^	26.93 ± 1.60 ^b^
	SBR-Rb	48.03 ± 1.72 ^a^	18.14 ± 0.12 ^b^	30.37 ± 1.35 ^a^
	SBR-W	45.40 ± 0.74 ^b^	19.58 ± 0.32 ^a^	30.14 ± 0.89 ^a^
50% Ethanol	SBR-B	41.28 ± 0.51 ^b^	19.75 ± 0.74 ^c^	36.78 ± 0.74 ^b^
	SBR-R	40.68 ± 0.91 ^b^	18.08 ± 0.14 ^d^	29.19 ± 1.63 ^d^
	SBR-Rb	44.08 ± 0.59 ^a^	20.93 ± 0.37 ^a^	32.39 ± 1.07 ^c^
	SBR-W	42.04 ± 0.20 ^b^	20.42 ± 0.20 ^ab^	41.88 ± 1.14 ^a^
Water	SBR-B	28.37 ± 0.66 ^b^	9.45 ± 0.13 ^a^	25.21 ± 1.55 ^a^
	SBR-R	25.65 ± 0.36 ^c^	7.60 ± 0.03 ^b^	18.51 ± 0.94 ^b^
	SBR-Rb	30.86 ± 0.26 ^a^	9.23 ± 0.36 ^a^	19.34 ± 1.34 ^b^
	SBR-W	30.18 ± 0.80 ^a^	9.52 ± 0.07 ^a^	25.51 ± 0.54 ^a^
Hot water	SBR-B	53.56 ± 0.64 ^a^	13.59 ± 0.44 ^a^	58.17 ± 0.21 ^b^
	SBR-R	46.72 ± 0.46 ^c^	11.05 ± 0.53 ^b^	53.31 ± 0.82 ^c^
	SBR-Rb	47.10 ± 0.86 ^c^	12.86 ± 0.33 ^a^	52.00 ± 0.68 ^c^
	SBR-W	49.89 ± 0.55 ^b^	13.24 ± 0.09 ^a^	59.95 ± 1.03 ^a^

Values are expressed as the mean ± standard deviation. n.d.: not detected. Means compared among the same solvent in the same column with different superscript letters are significantly different based on Tukey’s test (*p* < 0.05).

**Table 4 jof-07-00973-t004:** Tyrosinase-inhibitory activity of crude extracts prepared from solid-based residues (SBRs) using different solvents.

Conditions		Tyrosinase Inhibitory Activity (mg KAE/g Extract)	% Inhibition
Solvent	Residue		
Ethyl acetate	SBR-B	4.56 ± 0.19 ^c^	8.07 ± 0.33 ^c^
	SBR-R	5.14 ± 0.35 ^b^	9.08 ± 0.62 ^b^
	SBR-Rb	6.28 ± 0.16 ^a^	11.10 ± 0.28 ^a^
	SBR-W	5.03 ± 0.04 ^b^	8.89 ± 0.08 ^bc^
Acetone	SBR-B	9.01 ± 0.75 ^b^	15.93 ± 1.33 ^b^
	SBR-R	8.81 ± 0.28 ^b^	15.59 ± 0.50 ^b^
	SBR-Rb	10.12 ± 0.19 ^a^	17.90 ± 0.35 ^a^
	SBR-W	8.33 ± 0.32 ^b^	14.73 ± 0.53 ^b^
Ethanol	SBR-B	10.50 ± 1.09 ^a^	18.58 ± 1.92 ^a^
	SBR-R	9.78 ± 0.27 ^ab^	17.30 ± 0.47 ^ab^
	SBR-Rb	8.90 ± 0.09 ^b^	15.75 ± 0.17 ^b^
	SBR-W	10.44 ± 0.18 ^a^	18.47 ± 0.31 ^a^
50% Acetone	SBR-B	25.03 ± 1.01 ^b^	44.27 ± 1.78 ^b^
	SBR-R	22.42 ± 0.99 ^b^	39.66 ± 1.75 ^c^
	SBR-Rb	31.12 ± 1.00 ^a^	55.03 ± 1.76 ^a^
	SBR-W	23.73 ± 1.32 ^b^	41.98 ± 2.34 ^bc^
50% Ethanol	SBR-B	49.22 ± 0.18 ^b^	87.05 ± 0.31 ^ab^
	SBR-R	42.84 ± 0.66 ^c^	75.76 ± 1.18 ^b^
	SBR-Rb	51.13 ± 1.11 ^a^	90.43 ± 1.96 ^a^
	SBR-W	48.77 ± 0.79 ^b^	86.25 ± 1.41 ^c^
Water	SBR-B	10.52 ± 0.54 ^b^	18.61 ± 0.96 ^b^
	SBR-R	9.69 ± 0.65 ^bc^	17.14 ± 1.16 ^bc^
	SBR-Rb	11.75 ± 0.23 ^a^	20.79 ± 0.41 ^a^
	SBR-W	9.14 ± 0.67 ^c^	16.16 ± 1.19 ^c^
Hot water	SBR-B	22.12 ± 1.30 ^a^	39.13 ± 2.30 ^a^
	SBR-R	17.24 ± 0.56 ^b^	30.50 ± 1.00 ^b^
	SBR-Rb	20.67 ± 1.35 ^a^	36.56 ± 2.39 ^a^
	SBR-W	21.03 ± 1.06 ^a^	37.19 ± 1.87 ^a^

Values are expressed as the mean ± standard deviation. Means compared for the same solvent in the same column with different superscript letters are significantly different based on Tukey’s test (*p* < 0.05).

**Table 5 jof-07-00973-t005:** Pearson’s correlation coefficients among total phenolic content (TPC), total flavonoid content (TFC), total carbohydrate content (TCC), antioxidant activities, and tyrosinase-inhibitory activity.

Assay	DPPH	ABTS	FRAP	TYR
TPC	0.833 **	0.912 **	0.980 **	0.735 **
TFC	0.496 **	0.813 **	0.804 **	0.864 **
TCC	0.885 **	0.852 **	0.957 **	0.424 **

DPPH: DPPH radical scavenging activity; ABTS: ABTS radical scavenging activity; TYR: tyrosinase-inhibitory activity. **: Correlation is significant at *p* < 0.01 (two-tailed).

**Table 6 jof-07-00973-t006:** Phenolic compounds identified from 50% (*v*/*v*) ethanol and hot water extracts of SBR using LC–ESI-QqQ-MS/MS analysis.

Compounds	Equation	R^2^	Crude SBR Extracts (µg/g Extract)							
			EB	ER	ERb	EW	HB	HR	HRb	HW
*Phenolic acids*										
Gallic acid	y = 1.631.5x	0.9998	34.62 ± 1.92	17.65 ± 1.51	30.19 ± 2.60	40.63 ± 3.22	59.98 ± 3.32	28.90 ± 0.62	33.31 ± 1.88	48.50 ± 0.62
Protocatechuic acid	y = 2936.7x	0.9917	76.27 ± 1.81	40.23 ± 3.79	120.43 ± 3.31	124.26 ± 2.01	96.49 ± 5.58	37.31 ± 1.39	182.00 ± 2.73	175.66 ± 2.18
Chlorogenic acid	y = 3843x	0.9995	10.62 ± 1.17	15.69 ± 1.99	5.75 ± 0.86	7.21 ± 0.43	7.97 ± 0.76	20.94 ± 2.73	14.19 ± 1.28	10.31 ± 1.58
Caffeic acid	y = 5602.9x	0.9994	20.53 ± 2.71	17.07 ± 2.33	12.92 ± 0.99	19.07 ± 1.90	30.57 ± 1.03	15.23 ± 0.73	13.36 ± 0.60	41.83 ± 3.80
*p*-Coumaric acid	y = 3525.8x	0.9997	300.36 ± 5.40	306.51 ± 14.52	223.60 ± 11.96	258.24 ± 19.09	437.85 ± 36.64	364.16 ± 27.37	329.73 ± 15.75	315.15 ± 12.07
*o*-Coumaric acid	y = 3941.4x	0.9904	n.d.	n.d.	25.00 ± 1.90	n.d.	n.d.	n.d.	43.90 ± 2.07	n.d.
*p*-hydroxybenzoic acid	y = 2644.3x	0.9867	29.15 ± 0.53	52.95 ± 0.72	42.08 ± 0.42	18.31 ± 2.26	30.11 ± 0.85	55.37 ± 0.60	36.66 ± 0.32	25.04 ± 2.98
Total phenolic acid content			471.55 ± 6.45	450.10 ± 13.07	459.96 ± 7.94	467.72 ± 14.54	662.98 ± 37.39	521.91 ± 26.83	653.16 ± 15.43	616.50 ± 19.42
*Flavonoids*										
Rutin	y = 3711.3x	0.9996	n.d.	n.d.	151.96 ± 10.07	14.95 ± 1.28	n.d.	n.d.	44.86 ± 1.03	6.17 ± 0.46
Quercetin	y = 2161.6x	0.9996	n.d.	28.67 ± 1.29	71.67 ± 4.72	19.11 ± 0.27	n.d.	n.d.	n.d.	n.d.
Apigenin	y = 12889x	0.9912	63.61 ± 0.73	40.63 ± 3.13	26.92 ± 1.91	43.08 ± 5.95	22.96 ± 0.37	18.62 ± 0.39	18.77 ± 1.41	18.45 ± 0.77
Naringenin	y = 13145x	0.9893	68.48 ± 1.56	54.93 ± 3.36	64.44 ± 2.33	61.15 ± 1.28	51.71 ± 1.39	43.30 ± 2.23	52.87 ± 1.28	60.81 ± 4.81
Total flavonoid content			132.09 ± 0.83	124.23 ± 6.42	314.99 ± 11.75	138.30 ± 3.81	74.68 ± 1.32	61.93 ± 2.38	116.51 ± 2.65	85.43 ± 3.77

EB: 50% (*v*/*v*) ethanol extract of SBR-B; ER: 50% (*v*/*v*) ethanol extract of SBR-R; ERb: 50% (*v*/*v*) ethanol extract of SBR-Rb; EW: 50% (*v*/*v*) ethanol extract of SBR-W; HB: hot water extract of SBR-B; HR: hot water extract of SBR-R; HRb: hot water extract of SBR-Rb; HW: hot water extract of SBR-W. Values are expressed as the mean ± standard deviation of three replicates. n.d.: not detected.

**Table 7 jof-07-00973-t007:** Tentative identification of compounds detected in crude SBR extracts using LC-QTOF-MS/MS analysis.

No.	Tentative Compounds	Formula	Mass	RT (min)	M-H	M + H	Crude SBR Extracts							
							EB	ER	ERb	EW	HB	HR	HRb	HW
*Nucleosides*														
1	Adenosine 5′-monophosphate	C_10_H_14_N_5_O_7_P	347.0630	1.81	346.0550	348.0700	+	+	+	+	+	+	+	+
2	Adenosine	C_10_H_13_N_5_O_4_	267.0960	2.12		268.1030	+	+	+	+	+	+	+	+
3	Guanosine	C_10_H_13_N_5_O_5_	283.0917	2.79		284.0990							+	
4	Cordycepin	C_10_H_13_N_5_O_3_	251.1022	2.88		252.1095	+	+	+	+	+	+	+	+
*Nucleobases*														
5	Adenine	C_5_H_5_N_5_	135.0546	2.24	134.047	136.0618	+	+	+	+	+	+	+	+
*Amino acids*														
6	L-Lysine	C_6_H_14_N_2_O_2_	146.1060	1.56	145.0980	147.1130	+	+	+	+	+	+	+	+
7	L-Proline	C_5_H_9_NO_2_	115.0630	1.58	114.0560	116.0710	+	+	+	+	+	+	+	+
8	L-Arginine	C_6_H_14_N_4_O_2_	174.1117	1.60		175.1188	+	+	+	+	+	+	+	+
9	L-Glutamic acid	C_5_H_9_NO_4_	147.0530	1.95	146.0460	148.0610	+	+	+	+	+	+	+	+
10	Pyroglutamic acid	C_5_H_9_NO_3_	129.0426	2.42	128.0353						+			+
11	L-Isoleucine	C_6_H_13_NO_2_	131.0946	2.51		132.1018					+		+	
12	L-Tyrosine	C_9_H_11_NO_3_	181.0740	2.64	180.0670	182.0810	+	+	+	+	+	+	+	+
13	L-Leucine	C_6_H_13_NO_2_	131.0950	2.77	130.0870	132.1020	+	+	+	+	+	+	+	+
14	L-Phenylalanine	C_9_H_11_NO_2_	165.079	4.79	164.0718	166.0863	+	+	+	+	+	+	+	+
15	L-Tryptophan	C_11_H_12_N_2_O_2_	204.0900	7.53	203.0820	205.0970	+	+	+	+	+	+	+	+
*Peptides*														
16	Phe-Arg-Thr	C_19_H_30_N_6_O_5_	422.2166	3.64		423.2238								+
17	Val-Val	C_10_H_20_N_2_O_3_	216.147	4.09		217.1543				+	+	+		
18	PyroGlu-Pro	C_10_H_14_N_2_O_4_	226.0943	5.19		227.1025			+					+
19	Ser-Leu	C_9_H_18_N_2_O_4_	218.1265	5.48		219.1338		+					+	+
20	Gly-Ala-Ile	C_11_H_21_N_3_O_4_	259.1532	5.70		260.1605	+							
21	Asn-Leu	C_10_H_19_N_3_O_4_	245.1372	5.83		246.1445	+	+						+
22	Gly-Leu	C_8_H_16_N_2_O_3_	188.1161	5.85		189.1234	+	+	+			+		+
23	Alanyl-DL-leucine	C_9_H_18_N_2_O_3_	202.1316	5.90		203.1389		+	+		+			
24	Ile-Asp	C_10_H_18_N_2_O_5_	246.1211	6.24		247.1287			+					
25	Ile-Val	C_11_H_22_N_2_O_3_	230.1624	6.97		231.1696	+		+	+	+		+	+
26	Ser-Phe	C_12_H_16_N_2_O_4_	252.1104	7.16		253.1177	+			+				
27	Pro-Leu	C_11_H_20_N_2_O_3_	228.1469	7.17		229.1542	+	+		+	+	+	+	+
28	Gly-Phe	C_11_H_14_N_2_O_3_	222.0998	7.54		223.1069		+			+			
29	Ala-Phe	C_12_H_16_N_2_O_3_	236.1157	7.82		237.1229		+						
30	Leu-Leu	C_12_H_24_N_2_O_3_	244.1786	10.14		245.1859	+		+		+	+	+	
31	Val-Phe	C_14_H_20_N_2_O_3_	264.1473	10.19		265.1543							+	+
32	Ser-Ile-Leu	C_15_H_29_N_3_O_5_	331.2103	10.30		332.2175	+							
33	Val-Val-Thr	C_14_H_27_N_3_O_5_	317.1949	10.36		318.2022							+	
34	Ala-Val-Leu	C_14_H_27_N_3_O_4_	301.1997	10.53		302.2070					+			
35	Pro-Ile-Val	C_16_H_29_N_3_O_4_	327.2154	11.01		328.2227			+					
36	Val-Val-Ile	C_16_H_31_N_3_O_4_	329.2310	12.37		330.2384	+							
37	Leu-Val-Ile	C_17_H_33_N_3_O_4_	343.2464	14.16		344.2537	+							
*Sugars*														
38	D-Fructose 1-phosphate	C_6_H_13_O_9_P	260.0293	1.66	259.0220								+	
39	D-Fructose	C_6_H_12_O_6_	180.0630	1.75	179.0560		+	+	+					
40	Trehalose	C_12_H_22_O_11_	342.1157	1.79	341.1085		+	+	+	+	+	+	+	+
41	D-Mannitol	C_6_H_14_O_6_	182.0790	2.38	181.0720		+	+	+	+	+	+	+	+
*Phospholipids*														
42	Phytosphingosine	C_18_H_39_NO_3_	317.2929	19.60		318.3010	+	+	+	+	+	+	+	+
*Alkaloids*														
43	Trigonelline	C_7_H_7_NO_2_	137.0480	1.76		138.0550	+	+	+	+	+	+	+	+
44	Caffeine	C_8_H_10_N_4_O_2_	194.0810	9.45		195.0880			+		+	+		+
45	Nornicotine	C_9_H_12_N_2_	146.0843	21.84		147.0915				+	+	+		+
*Organic acids*														
46	L-Glyceric acid	C_3_H_6_O_4_	106.0270	1.84	105.0190		+	+	+	+	+	+	+	+
47	Malic acid	C_4_H_6_O_5_	134.0220	1.98	133.0140		+	+	+	+	+	+	+	+
48	Succinic acid	C_4_H_6_O_4_	118.0270	2.83	117.0190		+	+	+	+	+	+	+	
49	Azelaic acid	C_9_H_16_O_4_	188.1050	17.02	187.0970		+	+	+	+	+	+	+	+
*Vitamins*														
50	Niacinamide	C_6_H_6_N_2_O	122.048	2.21		145.035			+			+	+	+
51	Riboflavin	C_17_H_20_N_4_O_6_	376.136	9.04	375.1290		+	+	+	+	+		+	+
52	Thiamine	C_12_H_16_N_4_OS	264.1022	17.70	309.1005		+	+	+	+	+	+	+	+

EB: 50% (*v*/*v*) ethanol extract of SBR-B; ER: 50% (*v*/*v*) ethanol extract of SBR-R; ERb: 50% (*v*/*v*) ethanol extract of SBR-Rb; EW: 50% (*v*/*v*) ethanol extract of SBR-W; HB: hot water extract of SBR-B; HR: hot water extract of SBR-R; HRb: hot water extract of SBR-Rb; HW: hot water extract of SBR-W. RT: retention time.

## Data Availability

Not applicable.

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
