# Peer review of "The Feasibility of Utilizing Cultured Cordyceps militaris Residues in Cosmetics: Biological Activity Assessment of Their Crude Extracts"

_jof, 2021, doi:10.3390/jof7110973_

Round 1

Reviewer 1 Report

All the comments have been addressed. Now the manuscript is acceptable.

Author Response

Point : All the comments have been addressed. Now the manuscript is acceptable.

Response : Thank you very much for your acceptance.

Reviewer 2 Report

The authors have addressed all of my concerns previously raised about the article.

The paper is novel and interesting. The subject of the paper is of interest for the readers of the JoF. The conclusions are supported by the data. This is one profound study, at this is one of the most important strengths of the paper.

One of the weaknesses, according to my opinion, is abundant citations in the article. There are not inappropriate citations, just there are too many papers cited like this is a review paper and not the original research. Please try to address this issue as well, since the discussion part is too explanatory with familiar things. Just exclude some basic explanations and references pointing to them.

Taking in consideration all of the changes that authors made, I suggest the article for publication after minor revisions.

Author Response

Point : One of the weaknesses, according to my opinion, is abundant citations in the article. There are not inappropriate citations, just there are too many papers cited like this is a review paper and not the original research. Please try to address this issue as well, since the discussion part is too explanatory with familiar things. Just exclude some basic explanations and references pointing to them.

Response : Some references were considerably excluded from the manuscript, just the important ones are kept and used. Therefore, a total of 131 references are reduced to 83 ones (as seen in the revised manuscript file.

This manuscript is a resubmission of an earlier submission. The following is a list of the peer review reports and author responses from that submission.

Round 1

Reviewer 1 Report

  1. This manuscript studies cosmetics-related bioactivities of extracts from cultured Cordyceps militaris residues. The data look practical but the experimental design is doubtful particular the main compounds of the extracts were unknown.
  2. How many replicates were used in each extract? Basically, it needs at least three replicates and this part should be documented in the section of M&M.
  3. What are the main constitutes in each extract? Without such data, the authors could not describe the relationship between active compounds and their bioactivities.
  4. The iconic photos of the cells that were treated with extracts should be provided to demonstrate their growth, morphology, and viability, etc. According to my experiences, there is a risk to accept a paper that only presents quantitative data.
  5. Some minor errors in grammar in the text should be corrected.
  6. line 842: for further development - for the further development

Author Response

1. This manuscript studies cosmetics-related bioactivities of extracts from cultured Cordyceps militaris residues. The data look practical but the experimental design is doubtful particular the main compounds of the extracts were unknown.
This study was focused on extraction of phenolic compounds and polysaccharide in the residues of C. millitaris. They are mostly used as cosmetic active ingredients. To make it more clear, phenolic profile and composition were already proven by LC-MS/MS analysis.
2.How many replicates were used in each extract? Basically, it needs at least three replicates and this part should be documented in the section of M&M.
Three replicates were performed in all experiments. I already stated in material and methods, on the section of 2.17 (correlation and statistical analysis). All data can be also represented in term of mean±SD.
3. What are the main constitutes in each extract? Without such data, the authors could not describe the relationship between active compounds and their bioactivities.
Phenolic compounds and polysaccharides were mainly extracted and focused. Correlation analysis can estimate the relationship between these compounds and bioactivities. Our study was additionally conducted using LC-MS/MS analysis with both LC-QqQ-MS/MS and LC-QTOF-MS/MS techniques. The LC-QqQ-MS/MS was used to identify and quantify the phenolic compounds in the extracts. It was found that phenolic compounds seemed to be similar but their amounts were different in each crude extract. For LC-QTOF-MS/MS, it was done for screening and identifying other compounds found in the extracts. The detected compounds are already stated and compared with the previous study.

4. The iconic photos of the cells that were treated with extracts should be provided to demonstrate their growth, morphology, and viability, etc. According to my experiences, there is a risk to accept a paper that only presents quantitative data.
This study was done using MTT assay for determining the color absorption at 570 nm on microplate reader. This is a basic technique to indicate that the extracts are basically safe and allowed to use in cosmetic formulation.
5.Some minor errors in grammar in the text should be corrected.
The MS was re-checked and corrected in grammar.
6.line 842: for further development - for the further development
It was corrected and highlighted with the yellow color band (line 958).

Reviewer 2 Report

This study seems pretty interesting and the idea of the paper is very good. The introduction story goes well, material and methods together with results and discussions are clearly presented. However, since the proposition of the paper is for cosmeceutical application, the paper would need to present chemical composition studied by LC/MS.

For cosmeceutical application, it would be needed to standardize extracts to major compounds present . Therefore, I must suggest that for this type of application chemical analysis of the extracts is mandatory, regarding bioactive constituents. We need to know what are we dealing with. The manuscript has great potential, but according to my opinion this is issue must be addressed prior to publication. 

Author Response

This study seems pretty interesting and the idea of the paper is very good. The introduction story goes well, material and methods together with results and discussions are clearly presented. However, since the proposition of the paper is for cosmeceutical application, the paper would need to present chemical composition studied by LC/MS.
For cosmeceutical application, it would be needed to standardize extracts to major compounds present . Therefore, I must suggest that for this type of application chemical analysis of the extracts is mandatory, regarding bioactive constituents. We need to know what are we dealing with. The manuscript has great potential, but according to my opinion this is issue must be addressed prior to publication.

Response: This study was additionally conducted for determining chemical composition by LC-MS/MS analysis. Phenolic compounds which are our focused compounds were identified by LC-QqQ-MS/MS and found 7 phenolic acids and 4 flavonoids in the crude SBR extracts. The major phenolic compounds are already stated in the manuscript. In addition, LC-QTOF-MS/MS analysis was also carried out to screen and identified for other compounds. So the compounds which are identified can be used for standardization, and marker of the crude extracts can be addressed.